

2
# Variations in Boundary Layer Stability Across Antarctica: A Comparison Between Coastal and Interior Sites

3        Mckenzie J. Dice [1,2,3]

4        John J. Cassano [1,2,3]

5        Gina C. Jozef [1,2,3]

6        Mark Seefeldt [2,3]

7        [1] Department of Atmospheric and Oceanic Sciences, University of Colorado Boulder

8        [2] Cooperative Institute for Research in Environmental Sciences, University of Colorado Boulder

9        [3] National Snow and Ice Data Center, University of Colorado Boulder

**Key points**
● Self-organizing maps are used to examine the range of boundary layer stability profiles at two
12         continental and three coastal Antarctic sites.

● Near neutral to weak near surface stability usually occurs half or more than half of the time in all
14         seasons at the coastal sites but is infrequent at continental interior sites, except in the summer.

● When considering maximum stability near the surface or just above the boundary layer moderate
16         or stronger stability occurs almost always at the interior sites and often more than half of the time
17         at the coastal sites.

● At two of the three coastal sites analyzed here, moderate and strong stability occur more often
19         with clear than cloudy sky conditions at one of the coastal sites near neutral and weak stability
20         regimes occur more often with cloudy conditions.

*Correspondence to*: Mckenzie Dice, mckenzie.dice@colorado.edu



**Abstract.** The range of boundary layer stability profiles, from the surface to 500 m above ground level, present in radiosonde observations from two continental interior (South Pole and Dome Concordia) and three coastal (McMurdo, Georg von Neumayer III, and Syowa) Antarctic sites, is examined using the self-organizing maps (SOMs) neural network algorithm. A wide range of potential temperature profiles is revealed, from shallow boundary layers with strong near-surface stability to deeper boundary layers with weaker or near-neutral stability, as well as profiles with weaker near-surface stability and enhanced stability aloft, above the boundary layer. Boundary layer regimes were defined based on the range of profiles revealed by the SOM analysis. Twenty boundary layer regimes were identified to account for differences in stability near the surface as well as above the boundary layer. Strong, very strong, or extremely strong stability, with vertical potential temperature gradients of 5 to in excess of 30 K $(100 \text{ m})^{-1}$, occurred more than 80% of the time at South Pole and Dome Concordia in the winter. Weaker stability was found in the winter at the coastal sites, with moderate and strong stability (vertical potential temperature gradients of 1.75 to 15 K $(100 \text{ m})^{-1}$) occurring 70% to 85% of the time. Even in the summer, moderate and strong stability is found across all five sites, either immediately near the surface or aloft, just above the boundary layer. While the mean boundary layer height at the continental interior sites was found to be approximately 50 m, the mean boundary layer height at the costal sites was deeper, around 110 m. Further, a commonly described two stability regime system in the Arctic associated with clear or cloudy conditions was applied to the 20 boundary layer regimes identified in this study to understand if the two-regime behavior is also observed in the Antarctic. It was found that moderate and strong stability occur more often with clear than cloudy sky conditions, but weaker stability regimes occur almost equally for clear and cloudy conditions.



## 1 Introduction

Strong temperature inversions in Antarctica are the result of predominantly high albedo ice-covered surfaces and low sun angle in the summer and polar night in winter. All these factors contribute to prolonged surface radiative cooling which often results in statically stable boundary layers (King and Turner, 1997; Andreas et al., 2000) with temperature inversions sometimes exceeding 20 K (Lettau and Schwerdtfeger, 1967; Phillpot and Zillman, 1970). Increased solar radiation and warmer surface temperatures can result in near neutral or weakly stable conditions during the summer. Similar stability conditions can also occur at other times of year as a result of increased wind speeds or increased downwelling longwave radiation due to cloud cover (Hudson and Brandt, 2005; Stone and Kahl, 1991). This study aims to investigate the range of boundary layer stability that exists throughout the year at two continental interior sites and three coastal sites in Antarctica (Figure 1).

A previous study for McMurdo Station analyzed the range of boundary layer stability regimes present during the year-long Department of Energy (DOE) Atmospheric Radiation Measurement (ARM) West Antarctic Radiation Experiment (AWARE) campaign (Dice and Cassano, 2022). A strong seasonality of varying boundary layer stability was found, with the winter conditions dominated by strongly stable boundary layers (61% of the time), and summer conditions dominated by weak stability (83% of the time). Increased wind speeds in the winter were found to be responsible for reducing strong near-surface stability. This reduction of stability occurred near the surface while enhanced stability remained aloft in some cases. The results presented below aim to expand the analysis of boundary layer stability in Dice and Cassano (2022) to both continental and coastal locations across Antarctica.

Data from two additional coastal stations, Georg von Neumayer Station III (Neumayer Station) and Syowa Station (Figure 1) will also be analyzed, in addition to revisiting the data at McMurdo Station described above. Previously published results found surface-based temperature inversions occurred year-round at Neumayer Station, with a maximum frequency in the winter and a minimum in the summer, with 75% of the inversions having a strength of more than 1 K, and some up to 25 K, especially in the winter (König-Langlo and Loose, 2007; Silva et al., 2022). Some of the temperature profile structures observed by Silva et al. (2022) revealed multiple inversions within the same profile. This is similar to McMurdo Station where enhanced stability was often found to exist above a layer of weaker stability (Dice and Cassano, 2022). Cassano et al. (2016) found that stable boundary layer conditions occur 83% of the year over the northwestern Ross Ice Shelf (approximately 100 km from McMurdo Station), while neutral conditions occur 17% of the time. Further, 50% of the summer season was characterized by weakly unstable conditions, while stable stratification is dominant in the other three seasons (84% to 94%).

The continental interior of Antarctica is characterized by a short summer and a long, coreless winter (Hudson and Brandt, 2005). Stronger inversions and colder temperatures are often characteristic of higher elevation, continental interior sites (Phillpot and Zillman, 1970; Comiso, 1994; Zhang, et al., 2011), compared to coastal locations with weaker inversions and warmer temperatures (Phillpot and Zillman, 1970; Cassano et al., 2016). Continental interior sites also have greater inversion frequency than coastal sites, with inversion frequency in the fall and winter close to 100% (Zhang et al., 2011). At South Pole Station, inversions were found to be more common and stronger in the winter than in the summer. Hudson and Brandt (2005) also found inversions in the summer at Dome C to be stronger than those at South Pole. Inversions near the surface at Dome C can reach to 1 K m$^{-1}$ during polar night, and even stronger inversions, at 10 to 15 m above the surface, of up to 2.5 K m$^{-1}$ have been observed (Genthon et al., 2013).



Boundary layer stability in the polar regions in the winter has often been described as existing in
two distinct states (weak or strongly stable) driven by changes in cloud cover. The weakly stable regimes
occur under cloudy conditions, with increased downwelling longwave radiation warming the surface and
reducing stability. In contrast, clear sky conditions allow for strong radiative cooling and strong stability
(Stone and Kahl, 1991; Mahrt et al., 1998; Mahrt, 2014; Solomon et al., 2023). Stone and Kahl (1991)
described boundary layer stability at the South Pole as being in either a weakly stable or strongly stable
regime, associated with cloudy or clear conditions, respectively throughout the summer of 1986. Solomon
et al. (2023) distinguished between wintertime clear and cloudy regimes in the Arctic, during the
Multidisciplinary drifting Observatory for the Study of Arctic Climate (MOSAiC) campaign, to evaluate
model predictions of near-surface meteorological conditions including boundary layer stability. They
separated clear and cloudy regimes using the minima between the two peaks in the observed bimodal
probability distribution function (PDF) of net longwave radiation. Following Solomon et al. (2023) we
will identify clear and cloudy regimes based on the PDFs of net longwave radiation to determine if this
bimodal view of clouds, and associated boundary layer stability, found in the Arctic is also applicable to
coastal and interior sites across the Antarctic continent. We will also study how the clear-cloudy regimes
relate to the continuous range of stability regimes identified in this study.
This paper begins with a description of the observations from five Antarctic sites and details of
the methods used to analyze the data at these sites (Section 2). The results of this analysis will describe
the range and frequency of boundary layer stability profiles (up to 500 m AGL) at the sites (Section 3).
Additionally, differences in boundary layer stability associated with clear and cloudy conditions will be
presented. The results section will be followed by a discussion and comparison across coastal versus
continental interior locations (Section 4). A summary of these findings will follow, and the next steps in
this research will be identified (Section 5).

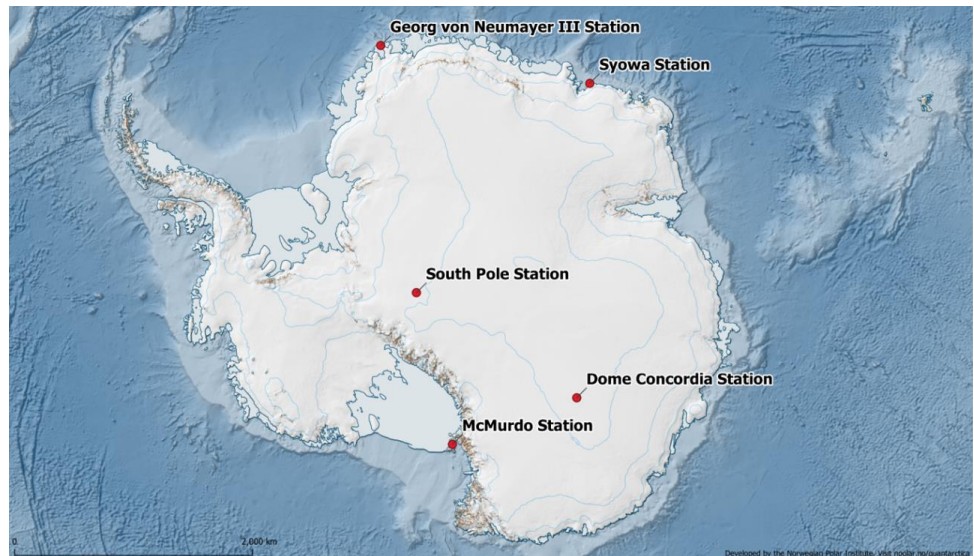

*Figure 1: Location of study sites (red dots with station names) across the Antarctic continent. Map*
*courtesy of Quantarctica (Matsuoka et al., 2018).*



## 2 Data and Methods

### 2.1 Data

The analysis presented in this paper is based on radiosonde and surface longwave radiation observations from three coastal (McMurdo Station, Neumayer Station, and Syowa Station) and two continental interior sites (South Pole Station, Dome Concordia Station) (Figure 1, Table 1, hereafter these five stations are referred to as McMurdo, Neumayer, Syowa, South Pole, and Dome C). The period of data used in the analysis at these five sites range from 13 months (McMurdo) to 19 years (Syowa). The differing time period evaluated at each site is due to varying amounts of time when radiosonde and radiation data are both continuously available, as well as periods of time when data was readily accessible. At McMurdo, this time period was chosen to coincide with availability of both radiosonde and radiation data from the AWARE campaign, which was previously analyzed by Dice and Cassano (2022). The Neumayer dataset is shorter than those at Syowa, Dome C, and South Pole, as Neumayer was not fully operational until 2009, and from 2009 to 2018, only 5 s temporal resolution radiosonde data was available. This data did not have sufficient vertical resolution for this study, thus only data after 2018 with 1 s temporal resolution was used. Syowa, Dome C, and South Pole all have longer continuous radiosonde and radiation datasets, that are easily accessible, lasting more than approximately 15 years.

South Pole is a high-elevation (2,835 m) continental interior site where strong surface inversions and extremely cold temperatures dominate (Zhang et al., 2011), and strong stability is almost constantly observed, especially in the winter (Phillpot and Zillman, 1970). The radiosonde data from the South Pole were retrieved from the Antarctic Meteorological Research and Data Center from 1 January 2005 to 29 September 2021. Radiosonde launches occur once daily at 2100 UTC for most of the year, with twice daily launches at approximately 0900 UTC and 2100 UTC during the short austral summer.

Dome C is another high-elevation (3,233 m) continental interior site characterized by cold temperatures and strong surface inversions, which occur throughout most of the year, and in the winter on a nearly permanent basis (Genthon et al., 2013; Pietroni et al., 2014, Vignon et al., 2017). The radiosonde data from Dome C are provided by the Antarctic Meteo-Climatological Observatory from 21 January 2006 to 14 October 2021. The radiosonde launches at Dome C are performed once daily at 1200 UTC year-round.

McMurdo is a coastal site located at the edge of the Ross Ice Shelf on the southwestern tip of Ross Island. The proximity of the Ross Ice Shelf, sea ice, open water, and the complex local topography near McMurdo results in a wide range of boundary layer stability types compared to the continental interior sites (Dice and Cassano, 2022). The McMurdo radiosonde data are from the DOE AWARE campaign (Lubin et al., 2017, 2020; Silber et al., 2018), which occurred at McMurdo from 20 November 2015 to 3 January 2017. The radiosonde launches during AWARE occurred twice per day at 1000 UTC and 2200 UTC.

Neumayer is near sea-level and located on the Ekström Ice Shelf, a relatively flat and homogeneous site. The meteorology and near-surface conditions are frequently influenced by large-scale cyclonic activity and sea ice fluctuations (Silva et al., 2022) resulting in changing boundary layer conditions. The radiosonde data from Neumayer are from the Baseline Surface Radiation Network (BSRN) from 1 June 2018 to 31 January 2021. Radiosonde launches occur once daily at approximately 1200 UTC, and twice daily during the summer months when conditions allow, at 0500 UTC and 1200 UTC.





Syowa is located on East Ongul Island in Lutzow-Holm Bay near sea level, with some low-
elevation slopes around it, and like the other coastal sites, it experiences warmer surface temperatures
compared to the continental interior. Syowa also experiences occasional strong wind due to katabatic flow
from the continental interior (Murakoshi, 1958). The radiosonde data from Syowa are from the Office of
Antarctic Observation Japan Meteorological Agency (pers. comm. Yutaka Ogawa) from 1 February 2001
to 23 January 2020. Radiosonde launches occur twice daily at 1130 and 2330 UTC.
Longwave radiation data were also obtained for all five sites to identify the clear and cloudy sky
conditions following the methods from Solomon et al. (2023). The radiation data are from the BSRN,
except at McMurdo where the data is from the AWARE campaign.
*Table 1: Information for each of the five study sites: South Pole, Dome C, McMurdo, Neumayer, and*
*Syowa. From left to right, the columns indicate: study site, coordinates and elevation above sea level*
*(ASL) of each site, site location type, the type of radiosonde and accuracy of the temperature and wind*
*measurements, respectively, the time period of the radiosonde launches, and the number of radiosonde*
*launches in the dataset.*

| Station | Coordinates, Elevation | Site Type | Instrument Type and Accuracy | Time Period of Radiosonde Launches | Number of Radiosonde Launch Profiles |
|---|---|---|---|---|---|
| South Pole | -89.98°S, 24.80°W; 2,836 m ASL | Interior plateau | Vaisala RS41-SGP radiosondes; 0.2 K, 0.5 m s$^{-1}$ | 01 Jan 2005-29 Sep 2021 | 8,587 |
| Dome Concordia | -75.10°S, 123.33°E; 3,251 m ASL | Interior plateau | RS-92 radiosondes; 0.2 K, 0.2 m s$^{-1}$ | 21 Jan 2006- 14 Oct 2021 | 5,147 |
| McMurdo | −77.85°S, 166.66° E; 10.1 m ASL | Coastal; Ross Island | RS-92 radiosondes; 0.2 K, 0.2 m s$^{-1}$ | 30 Nov 2015- 03 Jan 2017 | 786 |
| Georg von Neumayer | -70.65°S, -8.17°W; 38 m ASL | Coastal; Ekström Ice Shelf | Vaisala, RS41-SGP radiosondes; 0.2 K, 0.5 m s$^{-1}$ | 01 Jun 2018- 31 Jan 2021 | 1,220 |
| Syowa | -69.00°S, 39.58°W; 18.4 m ASL | Coastal; East Ongul Island | Meisei RS-11G radiosondes; 0.5 K, 2 m s$^{-1}$ | 01 Feb 2007- 23 Jan 2020 | 6,390 |



## 2.2 Methods

### 2.2.1 Self-Organizing Map

The goal of this paper is to analyze and compare the variability in boundary layer stability,
defined by potential temperature profiles, at five Antarctic research stations (Figure 1). Hundreds to
thousands of radiosonde profiles (Table 1), for each of the five sites, will be analyzed. The self-organizing
map, or SOM, algorithm is used to objectively identify patterns in the potential temperature profiles that
represent the range of conditions in the radiosonde observations.
The SOM algorithm is an unsupervised artificial neural network that groups similar patterns in
the training data into a user-specified number of patterns, which span the range of conditions in the
training data. The iterative training proceeds until the squared difference between the training data and the
SOM patterns is minimized (Kohonen et al., 1996; Hewitson and Crane, 2002; Cassano et al., 2015). The
resulting two-dimensional array of patterns is the master SOM, or simply the SOM. The SOM is
organized such that similar patterns are located adjacent to each other, while the most distinct patterns are
on opposite sides (Cassano et al., 2016). The SOMs presented here are trained using potential temperature
gradient profiles from the radiosonde observations. The potential temperature gradient profiles ($d\theta/dz$)
were used to train the SOM because this gradient defines the local static stability in the profile and allows
for classification of boundary layer stability regimes across seasons and sites. The SOMs in this study
were trained using the SOM-PAK software (http://www.cis.hut.fi/research/som-research), the details of
which are described by Kohonen et al. (1996).
The radiosonde data is interpolated onto a regular vertical grid before applying the SOM
algorithm, as described in Dice and Cassano (2022). Radiosonde profiles from all sites were interpolated
to a 5 m grid from 20 to 500 m above ground level. The lowest height of 20 m was selected since near-
surface warm biased temperatures are often present in radiosonde data observed below this height in
many profiles at the five study sites (Schwartz and Doswell, 1991; Mahesh et al., 1997). The top height of
500 m was chosen since this height encompasses the boundary layer features of interest.
To decide on the number of patterns to be identified by the SOM algorithm, several tests were
performed to find the appropriate SOM size to adequately represent the range of boundary layer profiles
present at each of the five sites. Unlike other iterative, unsupervised training algorithms, the SOM does
not identify distinct patterns, but a range of patterns which vary smoothly across the boundary layer states
observed in the radiosonde data. Identifying the proper SOM size is important for visualizing the full
range of boundary layer stability profiles present in the training data (Reusch et al., 2005; Cassano et al.,
2015). Too small of a SOM will result in important differences in the training data being lost in the few
generalized patterns, and too large of a SOM will be difficult to visualize, and only a few samples from
the training data may correspond, or "map" to each SOM pattern. Several SOM sizes were tested for this
analysis: 3 x 2 (6 patterns), 4 x 3 (12 patterns), 5 x 4 (20 patterns), 6 x 5 (30 patterns), and 7 x 6 (42
patterns). This initial evaluation of different SOM sizes found that a 6x5 SOM (Figures 2, 4, 6, 8, and 10)
best represented the boundary layer states present across the training data at all five sites. The 30 patterns
in the 6x5 SOM span the range of potential temperature profile types present in the training data, which
represents the hundreds to thousands of profiles (Table 1) from each of the five sites.
Once the SOM is trained, each individual radiosonde profile from the training data is "mapped"
to a single pattern in the SOM that it is most similar to by finding the pattern that has the smallest squared
difference between the radiosonde profile and the SOM-identified pattern. This mapping procedure
produces a list of best matching units, or BMUs, which identify the potential temperature gradient profiles
in the training data that correspond to each pattern in the SOM. Using this list, mean potential temperature





gradient and mean potential temperature anomaly (defined relative to the potential temperature at 500 m)
profiles are calculated and used to visualize the range of stability profiles present at each site (Figures 2,
4, 6, 8 and 10). The list of BMUs is also used to calculate the frequency of occurrence of each SOM
pattern and can be used to identify how boundary layer stability varies annually and seasonally. The
seasons are defined in this study as follows: summer (DJ), fall (FMA), winter (MJJA), and spring (SON).
These seasons are identified as such following previous definitions of Antarctic seasons (Cassano et al.,
2016, Nigro et al., 2017).
**2.2.2 Boundary Layer Regime Definitions**

The SOM analysis described above provides a relatively compact way to visualize the range of
boundary layer conditions present in the radiosonde observations, as well as their seasonality at the
various sites. However, this analysis does not allow for direct, quantitative comparison across the five
sites since unique SOMs are defined for each location. Thus, to compare the range of boundary layer
stability present at each of the five sites (Figure 1) the potential temperature gradient profiles are used to
define boundary layer stability regimes. The stability regime definitions are based on both the near-
surface stability (20 m to 50 m) and stability above the height of the boundary layer (up to 500 m) and
boundary layer depth.

Six near-surface stability regimes were defined (Table 2, left column) based on the potential
temperature gradient between 20 m and 50 m above ground, as this depth captures the near-surface
conditions while avoiding measurement errors below 20 m. The near-surface stability regimes range from
near neutral (NN; $d\theta/dz < 0.5$ K $(100 \text{ m})^{-1}$) to extremely strongly stable (ESS; $d\theta/dz > 30$ K $(100 \text{ m})^{-1}$).
Various thresholds to distinguish near neutral (NN), weak (WS), moderate (MS), strong (SS), very strong
(VSS), and extremely strong (ESS) stability were evaluated, and the thresholds listed in Table 2 were
found to best separate meaningful differences in near-surface stability across all five sites. These
thresholds were also evaluated, and found to be appropriate, in a separate study based on profiles
observed over Arctic sea ice as part of the MOSAiC expedition (Jozef et al. 2023).

It was also noted that many of the SOM patterns were characterized by a layer of stronger
stability above weaker stability near the surface, which was also noted by Dice and Cassano (2022) at
McMurdo. Therefore, the stability above the boundary layer is also used to define the overall stability
regime (Table 2). This requires identifying the top of the boundary layer, which is done following Jozef et
al. (2022) by using profiles of the bulk Richardson number. The bulk Richardson number is defined as the
ratio of buoyant turbulence production, or suppression, to mechanical generation of turbulence by wind
shear. A critical bulk Richardson number indicates the point at which turbulence cannot be sustained
(Stull, 1988). The boundary layer height is defined as the point in the profile where the bulk Richardson
number exceeds a critical value of 0.5 and remains above that critical value for at least 20 meters
consecutively.

Aloft stability regimes were determined with the same potential temperature gradient thresholds
as were used for the near surface stability regimes (Table 2). The maximum potential temperature
gradient above the boundary layer height and below 500 m was used to identify the aloft stability regimes.
Aloft-stability regimes were applied to any potential temperature gradient profile with a greater stability
aloft compared to the near-surface stability of that profile. No aloft stability regime is applied for cases
with the strongest stability near the surface.

Boundary layer stability regimes were also defined based on the depth of the boundary layer. In
analyzing all the boundary layer profiles it was found that there was a clear distinction between a group of
NN and WS regimes with boundary layer heights less than 125 m, and NN and WS regimes with





boundary layer heights much greater than 125 m. Thus, a very shallow mixed (VSM) stability regime was
defined to distinguish these cases, specifically for the NN and WS regimes with boundary layer depths
less than 125 m.

The near-surface and aloft stability regimes, along with the VSM regimes, were combined into an
overall stability regime, as listed in Table 3. For example, a profile identified as having near-neutral
stability near the surface with strong stability above the boundary layer, would be identified as near-
neutral, strong stability aloft, or NN-SSA. Thus, we end up with "stability groupings" with the same near
surface stability for multiple regimes, but with varying stability aloft. One example of these groupings is
the following: NN (near-neutral), NN-WSA (near-neutral, weak stability aloft), NN-MSA (near-neutral,
moderate stability aloft), and NN-SSA (near-neutral, strong stability aloft; Table 3).
*Table 2: Boundary Layer Regime definition scheme. The left column of the table shows the potential*
*temperature gradient (dθ/dz in K (100 m)$^{-1}$ thresholds used to define each of the six basic near-surface*
*stability regimes from 20 m to 50 m. The middle column shows how the very shallow mixed layer*
*definition was applied to NN and WS cases. The third column shows the maximum potential temperature*
*gradient thresholds (dθ/dz in K (100 m)$^{-1}$) for the aloft stability regimes.*

| Near-Surface Stability | Very Shallow Mixed Layer | Stability Above Boundary Layer ("Aloft") |
|---|---|---|
| **Near-Neutral (NN):** $d\theta \, dz^{-1} < 0.5$ K $(100 \, m)^{-1}$ | If near-surface stability = NN or WS and ABL height <125 m ➢ **Near-surface stability =Very-Shallow Mixed (VSM)** | |
| **Weak Stability (WS):** $d\theta \, dz^{-1} >= 0.5$ K $(100 \, m)^{-1}$ and $< 1.75$ K $(100 \, m)^{-1}$ | | **Weak Stability Aloft (-WSA):** $d\theta \, dz^{-1} >= 0.5$ K $(100 \, m)^{-1}$ and $< 1.75$ K $(100 \, m)^{-1}$ |
| **Moderate Stability (MS):** $d\theta \, dz^{-1} >= 1.75$ K $(100 \, m)^{-1}$ and $< 5$ K $(100 \, m)^{-1}$ | | **Moderate Stability Aloft (-MSA):** $d\theta \, dz^{-1} >= 1.75$ K $(100 \, m)^{-1}$ and $< 5$ K $(100 \, m)^{-1}$ |
| **Strong Stability (SS):** $d\theta \, dz^{-1} >= 5$ K $(100 \, m)^{-1}$ and $< 15$ K $(100 \, m)^{-1}$ | | **Strong Stability Aloft (-SSA):** $d\theta \, dz^{-1} >= 5$ K $(100 \, m)^{-1}$ |
| **Very Strong Stability (VSS):** $d\theta \, dz^{-1} >= 15$ K $(100 \, m)^{-1}$ and $< 30$ K $(100 \, m)^{-1}$ | | **Very Strong Stability Aloft (-VSSA):** $d\theta \, dz^{-1} >= 15$ K $(100 \, m)^{-1}$ |
| **Extremely Strong Stability (ESS):** $d\theta \, dz^{-1} >= 30$ K $(100 \, m)^{-1}$ | | **Extremely Strong Stability Aloft (-ESSA):** $d\theta \, dz^{-1} >= 30$ K $(100 \, m)^{-1}$ |

Regimes where no increased stability aloft is present (NN, WS, MS, SS, VSS, or ESS) as well as
the VSM-WSA will be referred to as "basic near-surface stability regimes". The reasoning for including
VSM-WSA in the basic near-surface stability regimes is that this regime is defined both by stability as
well as boundary layer depth. The VSM regime is derived from the same conditions that define the NN
and WS regimes, but in the VSM regime, a much shallower boundary layer exists (less than 125 m). The -



WSA in this regime is consistent with the potential temperature gradient that defines the VSM regime as a
whole and is thus considered as part of the basic near-surface stability regimes. Each stability grouping is
identified by a distinct color (Table 3): NN – brown; VSM – red; WS – green; MS – blue; SS – purple;
VSS – pink; ESS – indigo), in which the darkest color is the basic near-surface regime (no increased
stability aloft), and with decreasing color intensity as stability aloft in that regime grouping increases.
*Table 3: Boundary Layer Regime acronyms and color codes. On the left is the color and acronym used to*
*represent each of the 20 stability regimes in figures and tables throughout this paper, and the full regime*
*name is spelled out on the right. The basic near-surface stability regimes are denoted in bold font.*

| Regime Color and Acronym | Regime Full Name |
|---|---|
| **NN** | **Near Neutral** |
| NN-WSA | Near Neutral- Weak Stability Aloft |
| NN-MSA | Near Neutral- Moderate Stability Aloft |
| NN-SSA | Near Neutral- Strong Stability Aloft |
| **VSM-WSA** | **Very Shallow Mixed- Weak Stability Aloft** |
| VSM-MSA | Very Shallow Mixed- Moderate Stability Aloft |
| VSM-SSA | Very Shallow Mixed- Strong Stability Aloft |
| **WS** | **Weak Stability** |
| WS-MSA | Weak Stability- Moderate Stability Aloft |
| WS-SSA | Weak Stability- Strong Stability Aloft |
| **MS** | **Moderate Stability** |
| MS-SSA | Moderate Stability- Strong Stability Aloft |
| **MS-VSSA** | **Moderate Stability- Very Strong Stability Aloft** |
| MS-ESSA | Moderate Stability- Extremely Strong Stability Aloft |
| **SS** | **Strong Stability** |
| SS-VSSA | Strong Stability- Very Strong Stability Aloft |
| SS-ESSA | Strong Stability- Extremely Strong Stability Aloft |
| **VSS** | **Very Strong Stability** |
| VSS-ESSA | Very Strong Stability- Extremely Strong Stability Aloft |
| **ESS** | **Extremely Strong Stability** |



### 2.2.3 Clear and Cloudy Regime Classification

As mentioned in the Introduction, wintertime boundary layer stability in the polar regions is often described to be made up of two regimes, which differ based on the presence or absence of clouds and the associated differences in downwelling longwave radiation. This two-regime system is often defined as a "clear regime" with low values of downwelling longwave radiation, strong surface radiative cooling, and strong stability, and a "cloudy regime", with enhanced downwelling longwave radiation, surface warming and decreased near-surface stability (Phillpot & Zillman, 1970; Stone and Kahl, 1991; Solomon et al., 2023). Here, we will assess how the frequency of the 20 boundary layer regimes (Table 2) relate to the more commonly defined clear (strongly stable) and cloudy (weakly stable) regimes to evaluate the use of this more nuanced view of the relationship between boundary layer stability and cloud cover.

To determine the conditions with which the boundary layer regimes defined in Table 2 occur, we follow the approach of Solomon et al. (2023) that used net longwave radiation observations taken over the Arctic sea ice during the MOSAiC expedition to define clear and cloudy conditions. They found that during the winter there was a bimodal distribution of net longwave radiation. The minimum in frequency between the two peaks of this distribution was used to define clear and cloudy states, which were found to have distinct distributions of downwelling longwave radiation (Solomon et al. 2023). Following Solomon et al. (2023) this analysis will be completed only in the winter season.

PDFs of wintertime net longwave radiation are calculated at the five study sites (Figures S1 to S5) to determine if bimodal distributions of net longwave radiation are found at coastal and interior Antarctic sites, like what was found in the Arctic. Then, as in Solomon et al. (2023) we determine if distinct distributions in downwelling longwave radiation are present, which serve as a proxy for clear (small values of downwelling longwave radiation) or cloudy (large values of downwelling longwave radiation) conditions. Solomon et al. (2023) used the minima in the net longwave radiation PDF as a threshold to define clear and cloudy regimes. In this study, we define an overlap ratio (defined below) that quantifies how distinct the distributions of downwelling longwave radiation are for a given net longwave radiation threshold used to separate clear and cloudy states. For the identified net longwave radiation threshold, we create two PDFs of downwelling longwave radiation (Figures S1 to S5) based on the subset of observations corresponding to net longwave radiation values above (cloudy) or below (clear) the net longwave radiation threshold. Using the two downwelling longwave radiation PDFs, we determine the total number of clear cases, cloudy cases, and the number of coincident cases where the clear and cloudy PDFs overlap. The overlap ratio is calculated as the number of overlapping cases divided by the total number of clear and the total number of cloudy cases, and the final overlap ratio is the maximum of these two ratios. This overlap ratio quantifies how much overlap exists between the clear and cloudy downwelling longwave radiation PDFs and distinct clear and cloudy PDFs are characterized by low overlap ratios. The overlap ratio is calculated for each value of net longwave radiation (from the minimum to the maximum observed), at 1 W m$^{-2}$ intervals, at each site. The minimum overlap ratio at each site, from the calculations every 1 W m$^{-2}$, defines the net longwave radiation threshold identifying the most distinct distributions of downwelling longwave radiation for clear and cloudy cases. It generally corresponds to within a few W m$^{-2}$ of the minimum in bimodal PDF of net longwave radiation (vertical black line in Figures S1 to S5). The dates and times corresponding to the clear and cloudy states were used to determine the frequency of boundary layer stability regimes for the two states.



## 3 Results

### 3.1 South Pole

At a high-plateau, continental interior site such as South Pole, it is expected that strong stability will be present throughout much of the year (Phillpot and Zillman, 1970; Comiso, 1994; Hudson and Brandt, 2005; Zhang, et al., 2011). The SOM in Figure 2 shows the range of potential temperature profiles (anomaly and gradient) across 16 years of radiosonde observations at South Pole, as well as the stability regime (colored outline and label in top left of each pattern) corresponding to the mean profiles in each SOM pattern. The left side of the SOM is dominated by the strongest stability patterns, and stability decreases from left to right, with the weakest stability patterns in the upper right corner. Potential temperature gradients more than 5 K $(100\text{ m})^{-1}$ in nearly all of the SOM-identified patterns in Figure 2, with many greater than 15 K $(100\text{ m})^{-1}$, and some even greater than 30 K $(100\text{ m})^{-1}$, shows that strong stability is in fact common at this site. Potential temperature gradients in excess of 15 K $(100\text{ m})^{-1}$, corresponding to our VSS regime (Table 2), are rarely observed outside of the interior of Antarctica, even in the Arctic (Jozef et al., 2023). Potential temperature gradients less than 1.75 K $(100\text{ m})^{-1}$, corresponding to NN or WS regimes, occur only in patterns 6, 12, and 18 in the upper right of the SOM, emphasizing the dominance of strong stability at South Pole.

The height of the maximum potential temperature gradient within the profile varies across the SOM, often being located very close to the surface, as in the top left corner of the SOM, but sometimes the maximum gradient is located above a layer of decreased stability near the surface, as is in the bottom two rows of the SOM. These SOM patterns represent conditions with moderate or strong near-surface stability capped by enhanced stability aloft (-SSA, -VSSA, or -ESSA).

The SOM for South Pole (Figure 2) shows the boundary layer height for each SOM pattern, in addition to showing potential temperature gradient and anomaly profiles. The boundary layer depth rarely exceeds 100 m across the SOM and is very shallow (less than 50 m AGL) for the SS, VSS, and ESS cases present throughout much of the SOM. Boundary layer depth increases in the MS cases in the bottom right corner of the SOM (approximately 100 m) and is deepest in the NN and VSM cases in the top right of the SOM (just above 100 m).

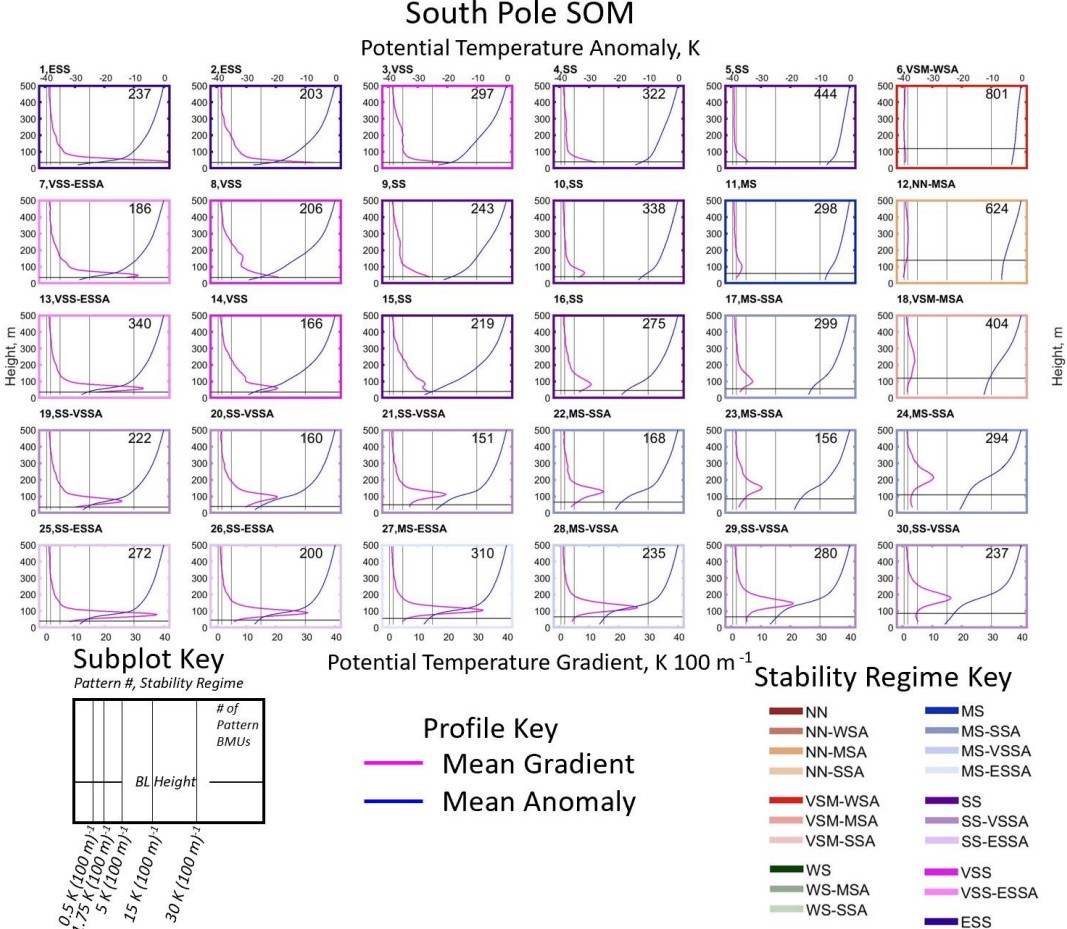

*Figure 2: Profiles of mean potential temperature gradient (pink line, top axis), and mean potential temperature*
*anomaly (blue line, bottom axis) calculated from the BMUs that map to each SOM pattern from 20 to 500 m above*
*ground level at South Pole.*

As mentioned in Section 2.2.2, the stability regime for each individual radiosonde profile was
identified to allow for comparison of regime frequencies across all five sites. Annual and seasonal
stability regime frequencies at South Pole are shown in Figure 3. When analyzing the frequency of
boundary layer stability regimes on an annual basis (Figure 3, left panel) the strongest near-surface
stability regimes (SS, VSS and ESS) are most common, occurring 58.5% of the time cumulatively. This
observation is consistent with what is seen in the SOM, where most of the profiles are SS, VSS, and ESS
regimes. For the weaker stability regimes (NN, VSM, and WS) the most common types of these regimes
are the ones with enhanced stability aloft indicating that most of the time when weak stability is present
near the surface moderate or strong stability remains aloft. Regardless of where strong stability occurs in
the profile (near-surface or aloft), strong stability, very strong stability, and extremely strong stability
occurs 85.1% of the time annually at the South Pole indicating that this location is dominated by the
strongest stability classes.



365    Seasonally there is a clear difference in regime frequencies between summer (DJ) and the other
366 three seasons. In the summer, the weakest near-surface stability regimes (NN and VSM) account for most
367 summer cases (58.7%), although often with enhanced stability aloft. Despite the sun being continuously
368 above the horizon during the summer, a high frequency of the MS and SS regimes (36.8%) still occurs.
369 WS regimes are very rare (4.5%), along with the VSS and ESS regimes, which almost never occur at this
370 time of year. In the winter (MJJA), SS, VSS, and ESS regimes dominate, occurring 68.9% of the time,
371 while NN and VSM occur only 10.7% of time, and WS and MS cases make up the remainder of stability
372 regimes observed in winter (3.4% and 16.9%, respectively). Interestingly, the few NN, VSM, and WS
373 cases in the winter all have strong stability aloft (-SSA), indicating that even when the weakest stability
374 regimes occur at the surface, strong stability is still present just above the boundary layer. The frequency
375 of stability regimes in the transition seasons (fall, FMA, and spring, SON) largely mirrors the frequency
376 of stability regimes in winter, again with the observation that the NN, VSM, and WS cases in the fall and
377 spring almost always have strong stability aloft (-SSA).

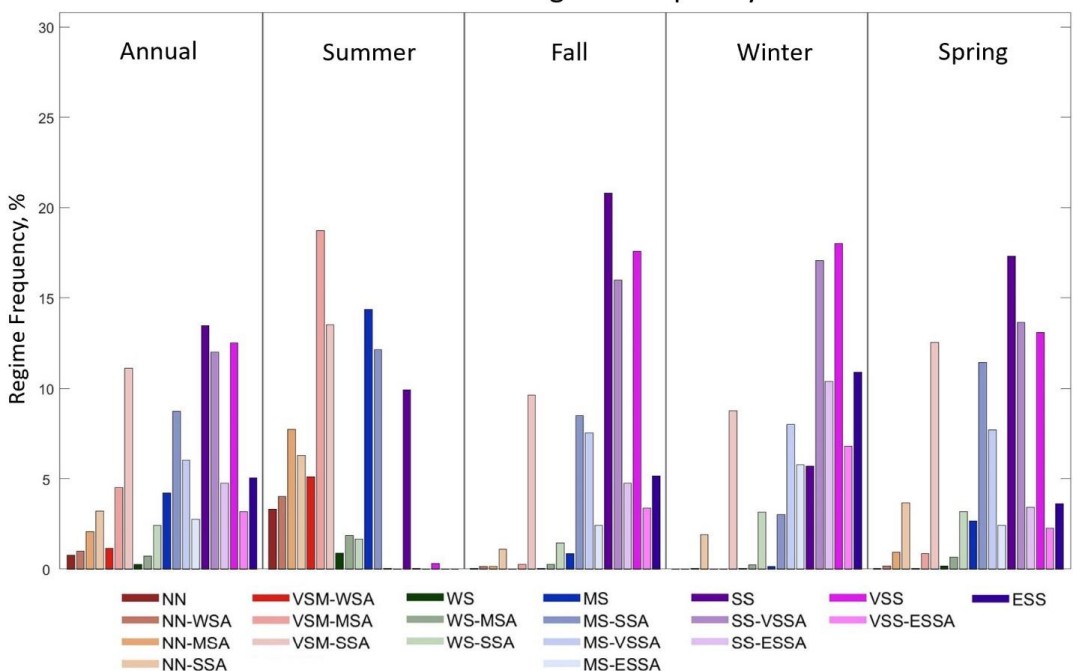

378 *Figure 3: Percentage of observations corresponding to each boundary layer stability regime observed at South Pole*
379 *annually (left panel) and seasonally (right 4 panels - summer, fall, winter, and spring). The regimes for the annual*
380 *and seasonal plots are arranged with increasing stability from left to right in each panel, and the order of the*
381 *stability regimes in each panel corresponds to the order of the regimes, from top to bottom and left to right in the*
382 *colored key at the bottom.*

383 **3.2 Dome C**

384    Dome C is another high-plateau continental interior site where strong stability persists throughout
385 much of the year (King and Turner, 1997; Andreas et al., 2000). This can be seen in the Dome C SOM in
386 Figure 4, where, like South Pole, most of the SOM-identified profiles exhibit potential temperature



gradients in excess of 5 K (100 m)$^{-1}$, and many are greater than 15 K (100 m)$^{-1}$. The left four columns of
the SOM are all SS or stronger stability regimes (greater than 5 K (100 m)$^{-1}$), and stability decreases from
left to right with the weakest stability patterns in the upper right corner (less than 1.75 K (100 m)$^{-1}$). The
height of the maximum potential temperature gradient within the profile changes across the SOM, with
the maximum stability observed at the surface in the upper left profiles, and the height of this maximum
stability increasing to the bottom right of the SOM, although the strongest stability usually occurs near the
surface in most of the SOM patterns.

The boundary layer height is less than 50 m across most of the SOM, and only increases when
stability decreases, such as in the bottom right, where stability is moderate and the boundary layer height
is about 75 m, and in the top right, where stability is weak, and the boundary layer height is around 100
m. In general, these are still very shallow boundary layers, even in the weaker stability patterns, compared
to other locations across the planet, where the height of the boundary layer can exceed 1000 m (Stull,
1988). Both at South Pole and Dome C strong, near-surface stability suppresses most of the mechanically
generated turbulence resulting in very shallow (typically less than 75 m) boundary layers. However,
shallow boundary layers at both sites also occur in the upper right portions of the SOM where relatively
weak stability exists, indicating that near-surface turbulent mixing is still confined to the lowest part of
the atmosphere (less than 150 m).

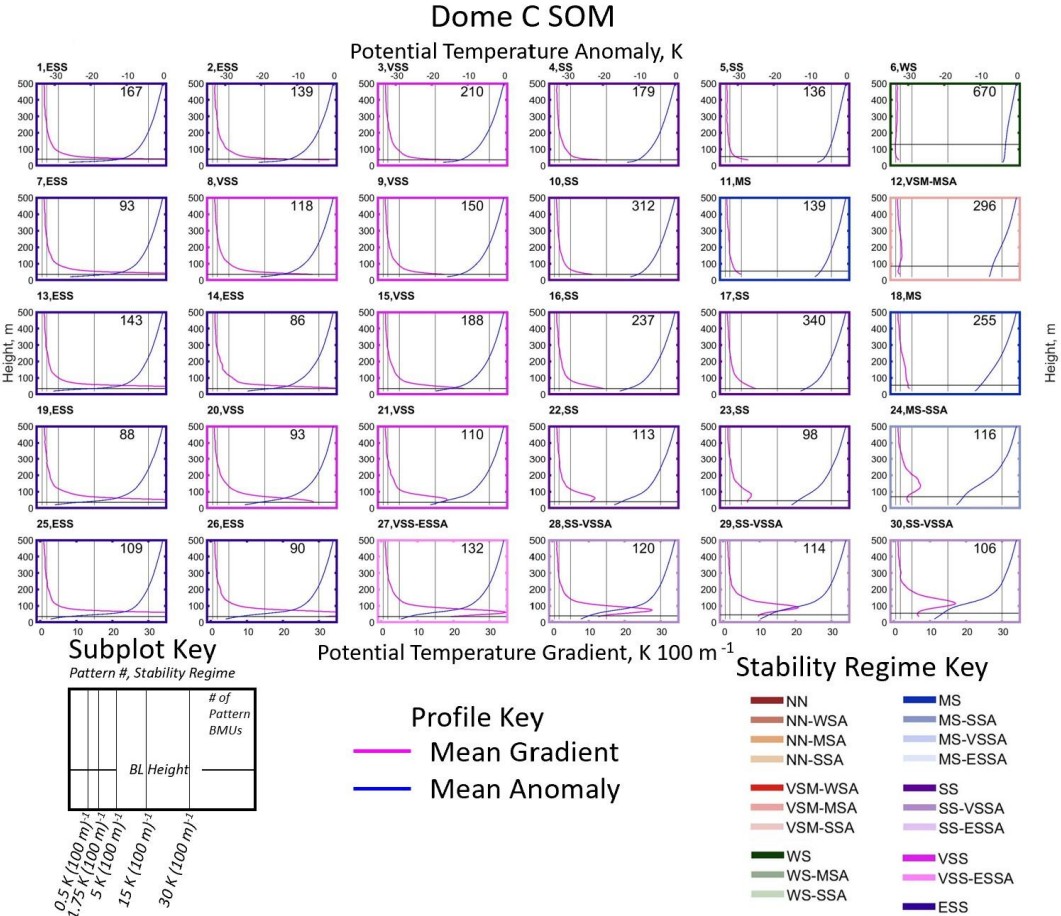

Figure 4: *Profiles of mean potential temperature gradient (pink line, top axis), and mean potential temperature anomaly (blue line, bottom axis) calculated from the BMUs that map to each SOM pattern from 20 to 500 m above ground level at Dome C.*

The frequency of occurrence of each stability regime at Dome C is shown in Figure 5. On an annual basis, SS, VSS and ESS regimes occur most frequently (73.6%), while the weaker stability regimes, NN, VSM, and WS only occur 13.5%. This is comparable to the range of stability regimes seen in the SOM, where these types of weaker stability regimes occur very rarely, and SS, VSS, and ESS regimes dominate across most of the SOM. A strong seasonal cycle emerges, with the weaker stability regimes dominant in summer and the strongest stability regimes dominant in winter. The summer season is largely characterized by NN, VSM, and WS regimes (61.4%), as well as MS regimes (31.1%). In the summer, SS, VSS, and ESS regimes occur only 7.5% of the time, indicating the rarity of strong stability at this time of year. In the winter, SS, VSS, and ESS regimes occur almost exclusively (96.7%), while all the other regime groupings (VSM, NN, WS, and MS) occur very rarely (3.3%). It is also interesting that the dominant regimes in the winter are solely the basic near-surface stability regimes of SS, VSS, and ESS regimes, and increased stability aloft in these regimes occurs much less frequently indicating that during the winter the strongest stability occurs at the surface most of the time, with infrequent cases of weakened stability near the surface and enhanced stability aloft. The frequency of stability regimes in the




transition seasons (fall and spring) is also dominated by stronger stability regimes (SS, VSS and ESS),
although with slightly lower frequencies than in winter, with these regimes occurring 83.7% and 76.9% of
the time in fall and spring respectively. The weakest stability regimes (VSM, NN, and WS) occur rarely
(4.6% and 4.9% of the time in fall and spring, respectively), while the MS regime occurs 11.7% and
15.7% of the time in fall and spring, respectively. In comparison to the summer and winter, the transition
seasons behave more like the winter season when it comes to regime frequency, with most regimes being
strong stability.

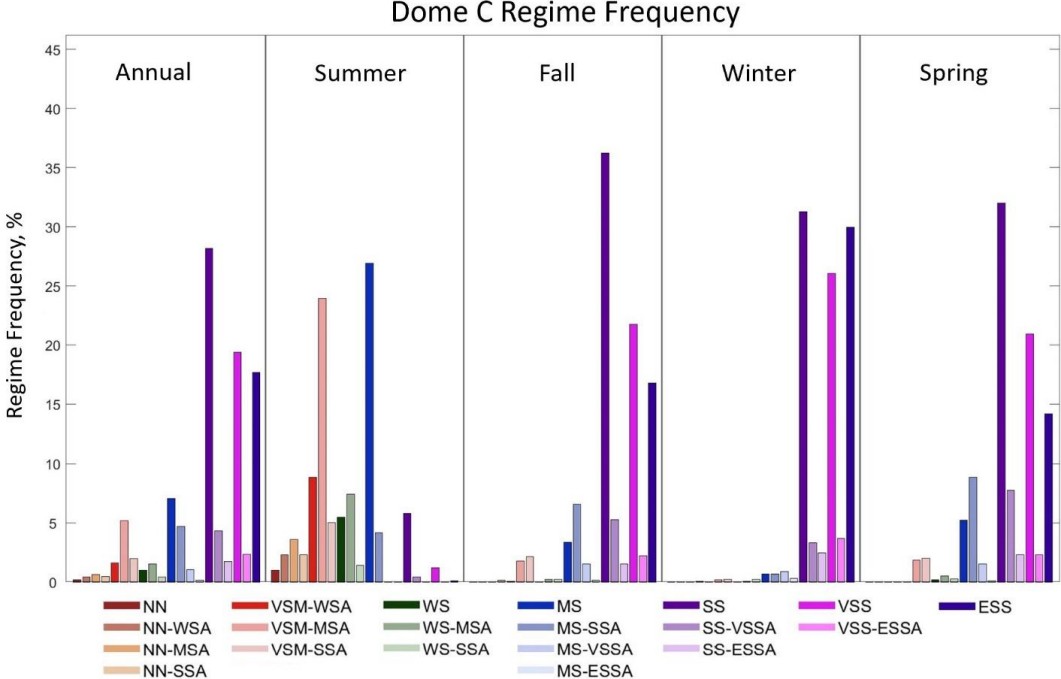

*Figure 5: Percentage of observations corresponding to each boundary layer stability regime observed at Dome C*
*annually (left panel) and seasonally (right 4 panels - summer, fall, winter, and spring). The regimes for the annual*
*and seasonal plots are arranged with increasing stability from left to right in each panel, and the order of the*
*stability regimes in each panel corresponds to the order of the regimes, from top to bottom and left to right in the*
*colored key at the bottom.*

### 3.3 McMurdo

So far, two continental interior sites, South Pole and Dome C have been analyzed, and now the
coastal sites, McMurdo, Neumayer, and Syowa will be analyzed. In comparison to the continental
interior, coastal locations are more exposed to the impacts of cyclonic activity, increased cloud cover and
moisture, as well as warmer surface temperatures and weaker inversions (Phillpot and Zillman, 1970;
Cassano et al., 2016). Given these previous observations, it is expected that weaker stability will be
present at the coastal sites compared to the near-constant state of strong stability observed at the colder,
continental interior sites described above.
Stability profiles at McMurdo identified by the SOM span a range from NN to SS regimes, as
seen in Figure 6. Stability in the SOM increases from left to right, with the weakest stability patterns in





the top left and strongest stability patterns in the bottom right. In addition to this gradient in stability
across the SOM the height of the strongest stability increases from the surface in the bottom rows of the
SOM to above a near surface layer of weaker stability in the top middle of the SOM. Most of these
patterns with enhanced stability aloft exhibit moderate or strong stability (-MSA or -SSA, respectively)
above a layer of weaker stability. Two-thirds of the SOM patterns exhibit potential temperature gradients
less than 1.75 K (100 m)$^{-1}$, corresponding to WS or weaker stability, and only five patterns on the right
side of the SOM (patterns 12, 18, 23, 24, and 30) exhibit strong stability with gradients greater than 5 K
(100 m)$^{-1}$. It can also be seen that the height of the boundary layer increases from the bottom right
(approximately 50 m) to the top left (approximately 200 m), as stability decreases, and the height of the
maximum stability increases in the profile.

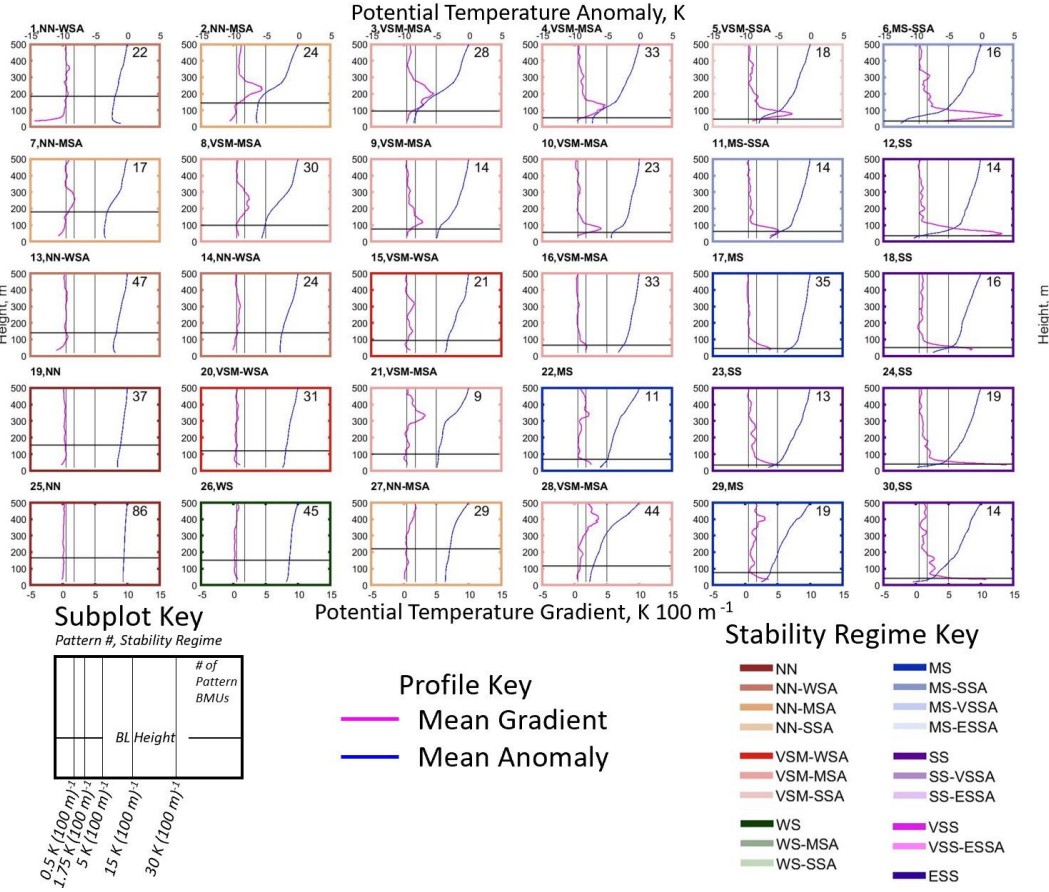

*Figure 6: Profiles of mean potential temperature gradient (pink line, top axis), and mean potential temperature*
*anomaly (blue line, bottom axis) calculated from the BMUs that map to each SOM pattern from 20 to 500 m above*
*ground level at McMurdo.*
Considering regime frequencies on an annual basis the NN and VSM regimes are most common
(68.6 %), followed by MS and SS regimes (24.9 %). The summer season is dominated by NN and VSM
regimes (91.2%), and WS, MS, and SS regimes occur only 8.8% of the time. This distribution of stability





is consistent with increased radiative forcing and previous observations of weaker stability in summer
compared to other seasons at a site approximately 100 km from McMurdo (Cassano et al., 2016). In the
winter, when it would be expected that strong stability would be dominant, only about half of the time
regimes with stability MS and greater occur (46.4%) while regimes with stability WS and weaker occur
just over half of the time (53.6%). However, when the regimes with stability WS and weaker occur,
moderate or strong stability aloft (-MSA and -SSA, respectively) is usually present (84% of NN, VSM,
and WS cases have -MSA or -SSA), indicating that even when weaker stability occurs near the surface
moderate or stronger stability is present just above the boundary layer. In the transition seasons, MS and
stronger cases occur 23.9% of the time in the fall and 28.2% of the time in the spring. NN and VSM cases
cumulatively occur 73.3% of the time in fall and 67.3% in the spring, while WS cases are largely absent.
In the VSM regime grouping, the -MSA and -SSA regimes are most common with the -WSA regime
occurring less frequently in comparison in both spring and fall. In the NN regime grouping, the frequency
of occurrence decreases with increasing stability aloft in the fall, and is more consistent across the -WSA,
-MSA, and -SSA regimes in the spring. This indicates that in the fall, it is more common for NN cases to
have weak rather than strong stability aloft, like what was observed in the summer, and opposite that in
the winter.

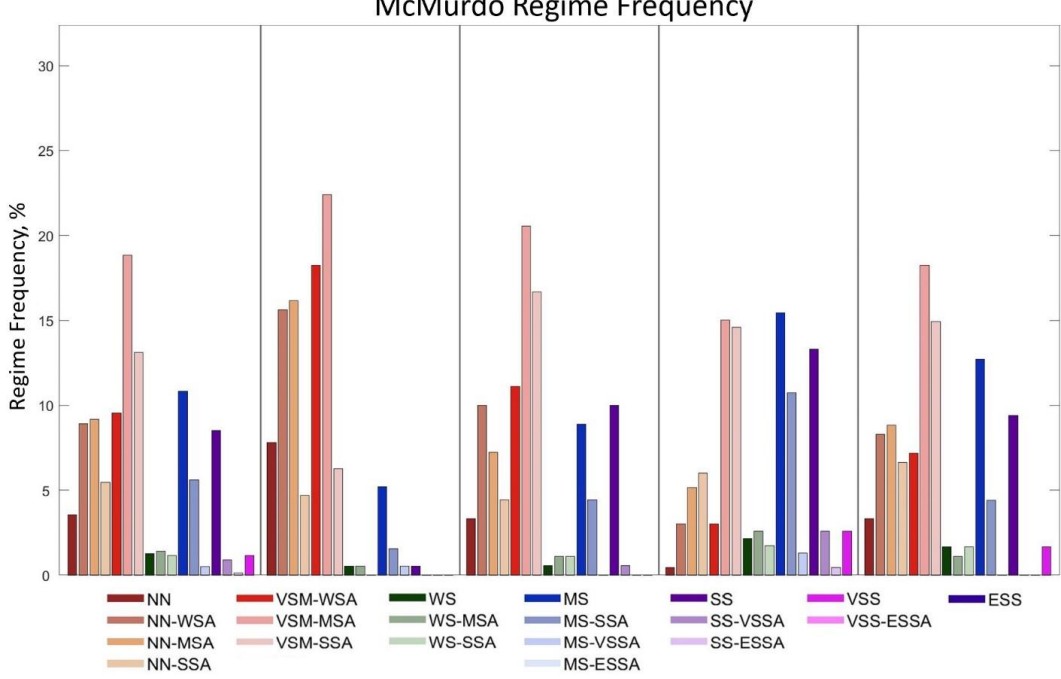

*Figure 7: Percentage of observations corresponding to each boundary layer stability regime observed at McMurdo*
*annually (left panel) and seasonally (right 4 panels - summer, fall, winter, and spring). The regimes for the annual*
*and seasonal plots are arranged with increasing stability from left to right in each panel, and the order of the*
*stability regimes in each panel corresponds to the order of the regimes, from top to bottom and left to right in the*
*colored key at the bottom.*





### 3.4 Neumayer

Neumayer is a coastal site located near sea-level, heavily influenced by large-scale cyclonic activity (Silva et al., 2022), and where the proximity of sea ice and open ocean can affect boundary layer stability throughout the year (Silva et al., 2022). Stability regimes at Neumayer span a range from NN to VSS regimes, as seen in the SOM in Figure 8. Generally, stability decreases from left to right across the SOM. Stability on the left side of the SOM decreases from the top to the bottom of the SOM, with the strongest stability regimes in the top left. On the right side of the SOM deep near neutral or weak stability patterns occur at the top of the SOM with patterns characterized by increasing stability aloft occurring towards the bottom of the SOM. This SOM shows two general modes of stability split by a bottom left to top right diagonal, with the portion to the right of this diagonal characterized by NN, VSM, and WS regimes, and the portion to the left characterized by MS, SS, and VSS regimes. The boundary layer height at Neumayer increases from the left side of the SOM, where very shallow boundary layers exist (less than 50 m) with strong stability, to the top right, where the boundary layer height increases to above 200 m.

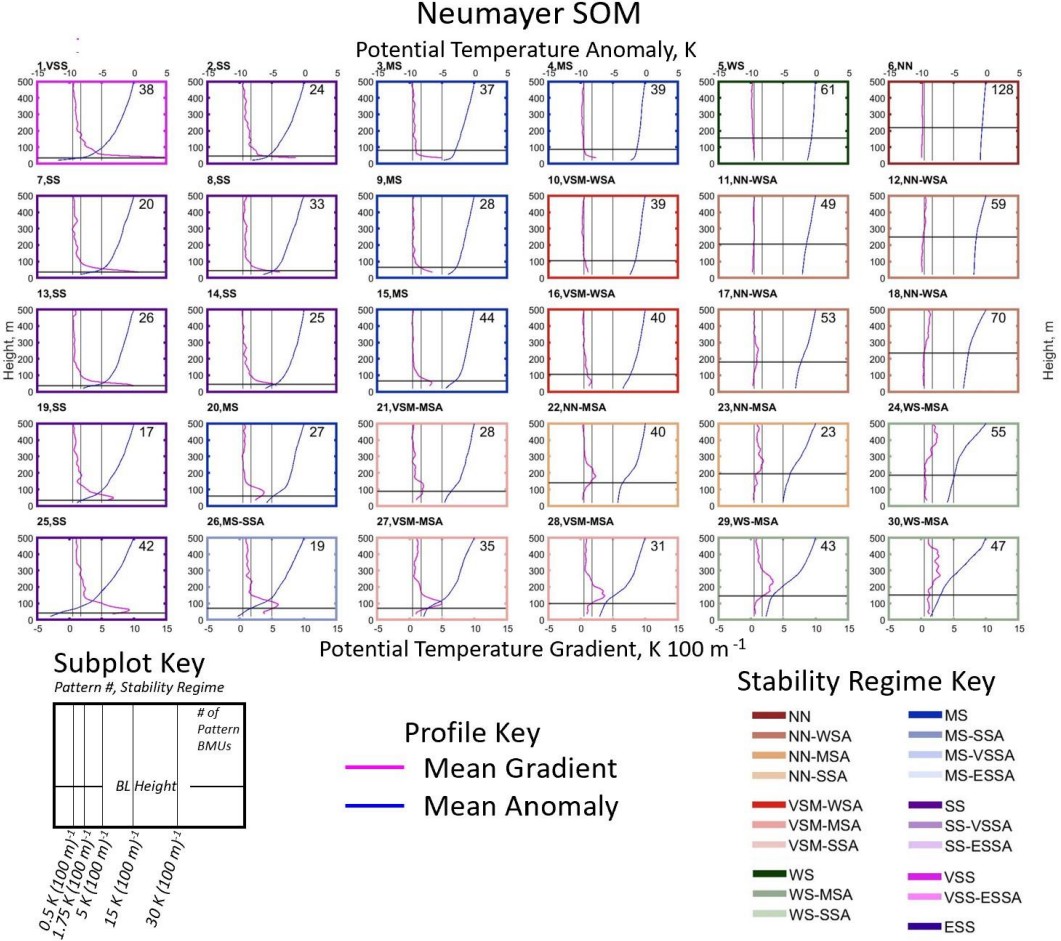

*Figure 8: Profiles of mean potential temperature gradient (pink line, top axis), and mean potential temperature*
*anomaly (blue line, bottom axis) calculated from the BMUs that map to each SOM pattern from 20 to 500 m above*
*ground level at Neumayer.*

On an annual basis, the NN and VSM regime groupings are most common (52.8%), and the MS
and SS (37.2%) regimes occur slightly less frequently at Neumayer (Figure 9). The WS regime grouping
occurs 8.4% while VSS and ESS regimes are rare and occur only 1.6% of the time throughout the year.
The summer season is dominated by NN and VSM regimes (74%). WS (6.1%), MS (14%), and SS
(5.9%) regimes are much less common in comparison. In the VSM and NN regime groupings regimes
with weak stability aloft (-WSA) are more common than those with stronger stability aloft (-MSA and -
SSA). In the winter, regimes with MS or greater stability are most common (60.1%), while regimes with
weaker stability, WS (12.2%), VSM (13%), and NN (14.7%), occur less frequently. Further, many of the
weaker stability regimes present in the winter are those with increased stability aloft, especially -MSA
and -SSA, indicating that moderate or stronger stability is frequently present either near the surface or
aloft in winter (89.5% of the time), whereas in the summer these moderate or strong stability cases (either
at the surface or aloft) cumulatively occur 50.7% of the time. In the fall, NN and VSM cases (47.9%) and
MS and stronger cases (44.6%) occur with almost equal frequency, unlike in the summer when the NN





and VSM cases are dominant, and winter when the MS and stronger cases are dominant. In the spring, the
VSM and NN cases (59.6%) occur more frequently than the MS and stronger cases (32.9%), which is
more similar to the distribution of regimes in the summer, when weaker stability regimes dominate.

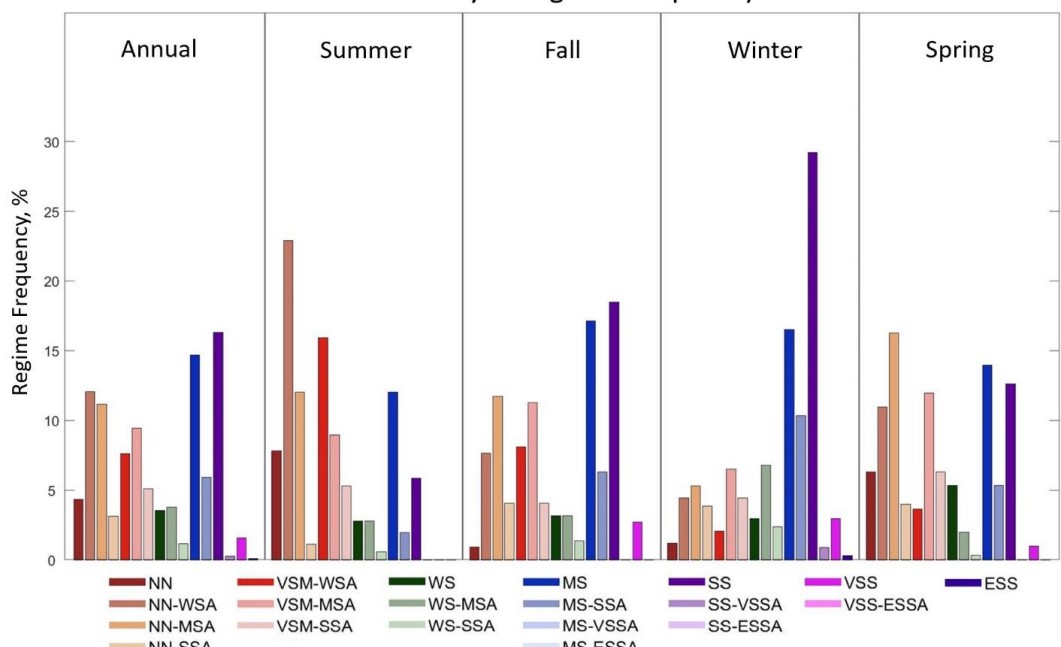

*Figure 9: Percentage of observations corresponding to each boundary layer stability regime observed at Neumayer*
*annually (left panel) and seasonally (right 4 panels - summer, fall, winter, and spring). The regimes for the annual*
*and seasonal plots are arranged with increasing stability from left to right in each panel, and the order of the*
*stability regimes in each panel corresponds to the order of the regimes, from top to bottom and left to right in the*
*colored key at the bottom.*
**3.5 Syowa**
Syowa is a coastal site near sea-level, impacted cyclonic activity and by katabatic winds from the
continental interior (Murakoshi, 1958), which sometimes result in strong wind events (Yamada and
Hirasawa, 2018). Stability at Syowa spans a range from NN (top left corner of SOM) to SS (bottom right
corner of SOM) regimes, as seen in the SOM in Figure 10. Stability generally increases from left to right
and top to bottom across the SOM. The height of the maximum potential temperature gradient is near the
surface on the far-right side of the SOM and increases to approximately 300 m in the bottom left. Shallow
boundary layers associated with the strong stability patterns in the bottom right increase in height to the
top left, where near-neutral conditions extend through a deeper, 200 m boundary layer.



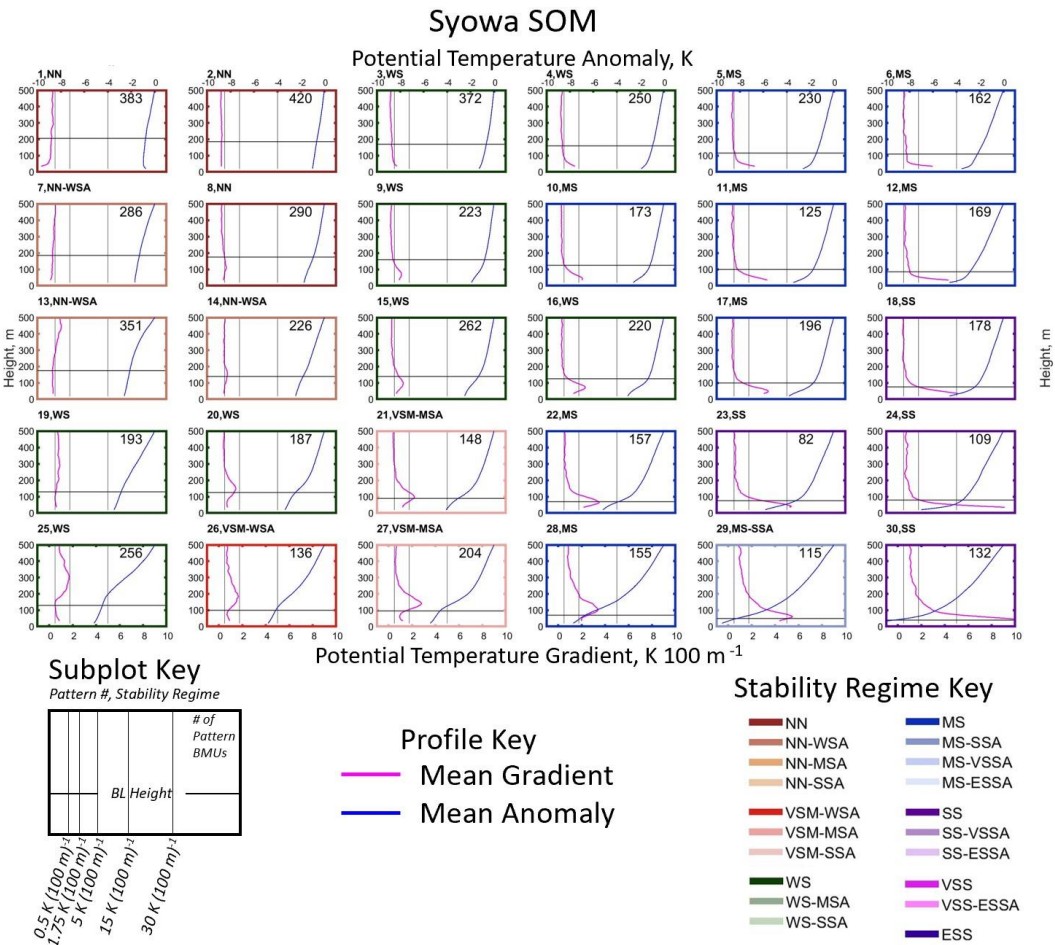

*Figure 10: Profiles of mean potential temperature gradient (pink line, top axis), and mean potential temperature anomaly (blue line, bottom axis) calculated from the BMUs that map to each SOM pattern from 20 to 500 m above ground level at Syowa.*

The frequency of occurrence of each stability regime at Syowa, annually and seasonally, is shown in Figure 11. On an annual basis, a mix of regimes are observed, mostly in the NN (21.8%) and VSM (31.4%) regime groupings, with enhanced stability aloft common. The WS regime (15.1%) and MS regime (25.1%) also occur frequently, on an annual basis, but enhanced stability aloft rarely occurs in these regime groupings. The strongest stability regimes (SS, VSS and ESS) occur infrequently (6.8%). These results indicate that near neutral to moderate stability is most common at Syowa, while stronger stability is rare. The summer season is dominated by the NN and VSM regimes (71.5%), while the WS regime occurs 11.4% of the time, and the MS regime 15.3% of the time. In all regime groupings in the summer, strong stability aloft (-SSA) regimes are less common than weak or moderate stability aloft (-WSA and -MSA, respectively), which is reflective of the lack of strong stability regimes in general in this season. In the winter, MS and SS regimes (45.4%) occur about as often as the NN and WS regimes (43.4%), but MS is by far the most common individual regime in winter (31%). Regimes with increased stability aloft (-MSA and -SSA) are uncommon in the winter except in the VSM regime grouping, and





rather the basic near-surface stability regimes (without enhanced stability aloft) or -WSA cases are more
common. In the transition seasons, a variety of regimes occur with similar frequencies. In the fall the most
common regime groupings are the VSM cases (34%) followed by the NN cases (24.2%) and the MS cases
(20.4%), and in the spring, the VSM (30.5%) regimes are most common followed by MS (24.6%), and
NN (24.5%) regimes that occur with nearly identical frequencies. In both seasons, like the summer and
winter, -MSA and -SSA cases occur rarely, with -WSA being more common when increased stability
aloft is observed for a given regime grouping.

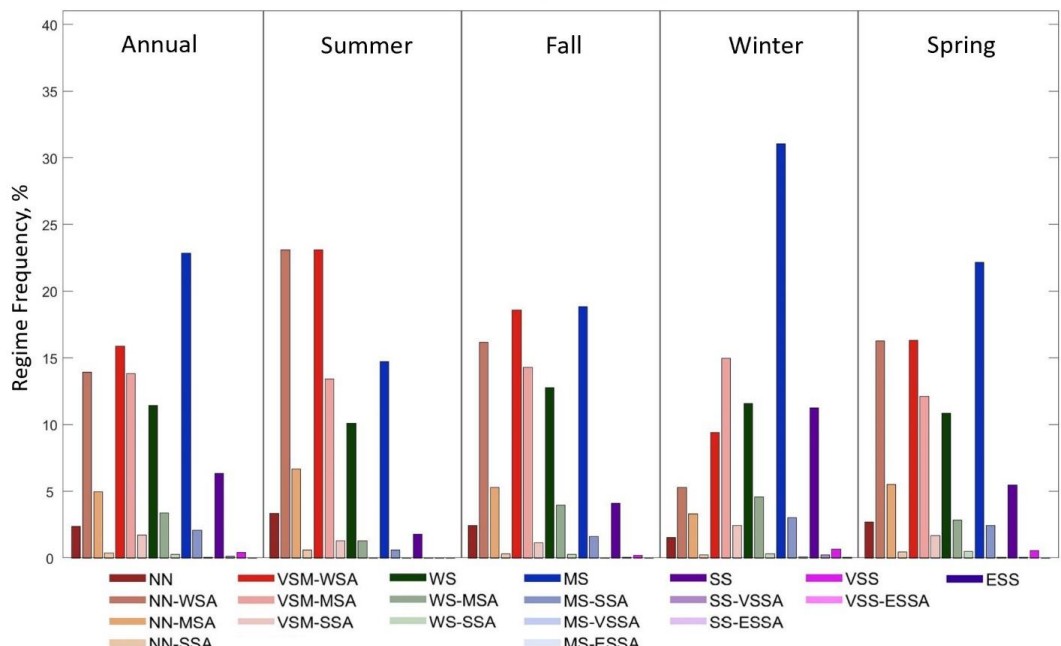

*Figure 11: Percentage of observations corresponding to each boundary layer stability regime observed at Syowa*
*annually (left panel) and seasonally (right 4 panels - summer, fall, winter, and spring). The regimes for the annual*
*and seasonal plots are arranged with increasing stability from left to right in each panel, and the order of the*
*stability regimes in each panel corresponds to the order of the regimes, from top to bottom and left to right in the*
*colored key at the bottom.*

**3.6 Stability Regime Frequencies for Clear and Cloudy Conditions**

As discussed in the introduction and methods, stability in the polar boundary layer is often
described in the literature as a two-regime system, with cloudy states characterized by large values of
downwelling longwave radiation and weak stability and clear states characterized by small values of
downwelling longwave radiation and strong stability (Mahrt et al., 1998; Mahrt, 2014; Solomon et al.,
2023). To determine if this two-regime description of boundary layer stability and cloud cover is observed
in the Antarctic a clear or cloudy attribution was given to each radiosonde profile based on the surface net
longwave radiation value at the time of launch following the method described in Section 2.2.3, based on
Solomon et al. (2023).



Solomon et al. (2023) found that the difference between cloudy and clear states in the Arctic
could be defined by a threshold value of net longwave radiation marking the minimum in the PDF
between two peaks in a bimodal distribution of net longwave radiation. PDFs of winter net longwave
radiation at the five Antarctic sites analyzed in this paper are shown in Figures S1 to S5. The PDFs for the
two interior sites (Dome C and South Pole, Figures S1 and S2) do not show a bimodal distribution while
the three coastal sites do (Figures S3 to S5). The overlap ratio for the cloudy and clear downwelling
longwave radiation PDFs for each site, as described in Section 2.2.3, further support the lack of distinct
cloudy and clear radiative states at the interior sites, with large values of this ratio (0.84 at South Pole and
0.91 at Dome C) indicating that there is no value of net longwave radiation that allows a meaningful
separation between cloudy and clear states with unique distributions of downwelling longwave radiation.
In contrast, the three coastal sites have overlap ratios of less than 0.5 (0.19 for McMurdo, 0.33 for
Neumayer, and 0.46 for Syowa) for net longwave radiation threshold values that correspond closely to the
minimum in the net longwave radiation PDF (Figures S3 to S5), indicating that distinct downwelling
longwave radiation distributions exist for cloudy and clear states at these sites. As such, we will evaluate
the frequency of stability regimes for cloudy and clear conditions at the three coastal sites, but not for the
interior sites.

Figure 12 shows the frequency of each stability regime for cloudy (solid bars) and clear (hatched
bars) cases for the three coastal sites: McMurdo, Neumayer, and Syowa. At McMurdo (Figure 12a), the
most obvious result is that in the MS, SS, and VSS regimes occur much more frequently during the clear
sky state. This result is consistent with previous observations that clear skies allow for radiative cooling
and the development of strong near surface stability (Stone and Kahl, 1991; Hudson and Brandt, 2005). In
contrast, the NN and WS regimes generally occur preferentially during cloudy conditions, also consistent
with previous results that increased cloud cover reduces near-surface stability (Stone and Kahl, 1991;
Hudson and Brandt, 2005). Interestingly, the VSM and NN-SSA regimes occur nearly equally regardless
of cloud cover. This indicates that changes in downwelling longwave radiation related to varying cloud
cover do not play a dominant role in the forcing of these regimes.

When examining the distribution for Neumayer (Figure 12b), the SS regime is over twice as
frequent during clear compared to cloudy conditions, as expected (Stone and Kahl, 1991; Mahrt et al.,
1998; Mahrt, 2014; Solomon et al., 2023). The same is true for the VSS regime, and clear conditions are
present for the singular ESS regime as well. The MS and MS-SSA regimes also occur more frequently
with clear rather than cloudy conditions. The NN regimes usually occur with cloudy compared to clear
conditions. The various VSM and WS regimes have occurrences where sometimes clear, and sometimes
cloudy, periods are dominant. There are also VSM and WS regimes where they are roughly equal. This
suggests that changes in downwelling longwave associated with changes in cloud cover do not play a
primary role in forcing the VSM or WS regimes to occur.

Finally, at Syowa (Figure 12c), an interesting pattern emerges, where the frequency of most
stability regimes is similar for both cloudy and clear conditions. This is surprising, given that previous
studies have found weaker stability is favored by cloudy conditions, and stronger stability is favored by
clear conditions. This is not the case at Syowa, and may indicate that changes in downwelling longwave
radiation, associated with cloudy and clear conditions, do not exert a strong control on near surface
stability at this site.



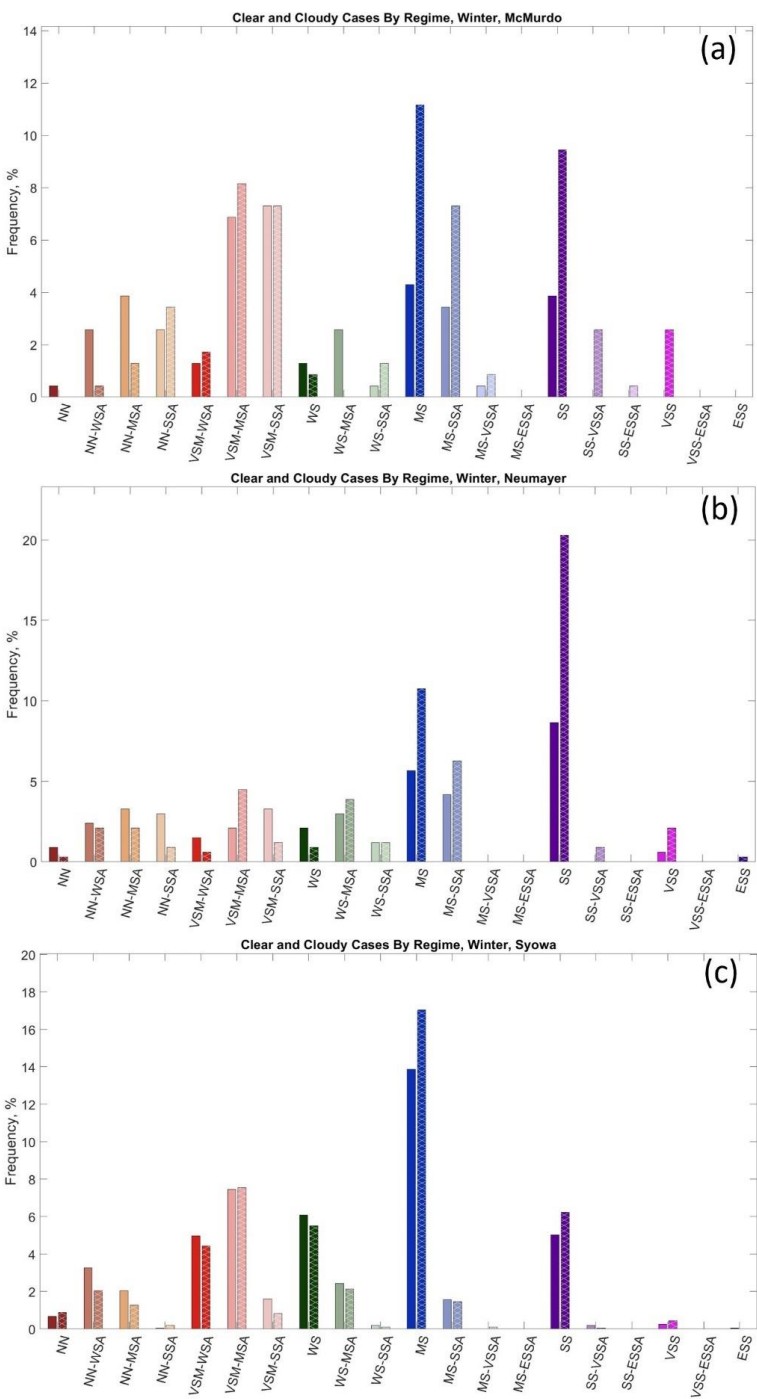

*Figure 12: The distribution of the various boundary layer stability regimes at McMurdo (a), Neumayer (b), and*
*Syowa (c) split into cloudy (left, solid bars) clear (right, hatched) observations in the winter season.*



**4 Discussion and Conclusions**

SOMs have been used in the results presented above to identify the range of boundary layer stability profiles at two continental interior and three coastal Antarctic sites (Figures 2, 4, 6, 8 and 10). Based on the SOM analysis a quantitative boundary layer stability definition was developed and applied to classify the SOM patterns into unique stability regimes. While several studies have examined general trends in boundary layer stability at individual sites in Antarctica (Hudson and Brandt, 2005; Cassano et al., 2016; Silva et al., 2022), or estimated inversion strength empirically (Philpot and Zillman, 1970), no known study has completed a widespread comparison of the range and seasonality of boundary layer stability across the continent.

The stability regimes present, and frequency of these regimes, differed between the continental interior sites and the coastal sites. At the interior sites, South Pole and Dome C, strong stability patterns dominate the SOM consistent with previous studies of near-surface stability on the polar plateau (Hudson and Brandt, 2005; King and Turner, 1997; Andreas et al., 2000). Twenty seven of 30 patterns at South Pole (Figure 2) and 28 of 30 at Dome C (Figure 4) have stability between MS and ESS, with potential temperature gradients in excess of 30 K (100 m)$^{-1}$ in several of the SOM profiles. Some of the SOM-identified profiles at these sites have weaker stability near the surface, with stronger stability aloft, and these patterns are more common at South Pole (Figure 2, bottom two rows) than at Dome C (Figure 4, bottom right corner). Finally, there are generally more VSS and ESS patterns in the Dome C SOM (left two columns) compared to the South Pole SOM (upper left corner), indicating stronger stability at this site, which was also observed by Hudson and Brandt (2005).

In contrast to the interior sites, at the coastal sites, McMurdo (Figure 6), Neumayer (Figure 8), and Syowa (Figure 10), the SOM profiles are more evenly distributed across NN, VSM, WS, MS, and SS profiles with only one VSS profile and no ESS profiles. Across all three coastal sites, over half of the SOM-identified patterns have a potential temperature gradient less than 1.75 K (100 m)$^{-1}$. These gradients occurred for only two or three patterns at Dome C and South Pole, respectively. This indicates more favorable conditions for weaker near-surface stability at coastal sites (Phillpot and Zillman, 1970; Cassano et al., 2016). This clearly distinguishes the boundary layer conditions of the continental interior sites from those at the coastal sites, as also noted by Lettau and Schwerdtfeger, (1967), Phillpot and Zillman, (1970), Comiso, (1994), Zhang, et al. (2011), and Cassano et al. (2016). It is also important to note the common occurrence of enhanced stability above a layer of weaker near-surface stability in the SOMs for the coastal sites in comparison to the continental interior sites. This phenomenon rarely occurs in the Dome C SOM, only in the bottom right corner (Figure 4), and across the bottom two rows in the South Pole SOM (Figure 2), but across much of the SOMs for McMurdo (Figure 6) and Neumayer (Figure 8), and some of the SOM for Syowa as well (Figure 10).

The SOM analysis indicates a mean boundary layer depth being much shallower at Dome C (45 m) and South Pole (60 m) compared to the coastal sites (95 m to 120 m). The strong near-surface stability that is almost always present at the continental interior sites limits the depth and strength of turbulent mixing, while weaker stability at the coastal sites allows for stronger near-surface turbulence and thus increased boundary layer depths. This behavior of boundary layer depth is also observed by King and Turner (1997), who found shallow boundary layers in the continental interior with boundary layer depth increasing towards the coasts. Pietroni et al. (2012) estimated the wintertime boundary layer height at Dome C using the bulk Richardson number and found it to be always below 150 m, but usually less than 50 m, and Aristidi et al. (2005) found shallower boundary layer depths at Dome C (less than 50 m) compared to South Pole, consistent with our results.





To further summarize and compare the frequency of occurrence of boundary layer regimes
(defined in Table 2) across the Antarctic continent, Figure 13 and Table S1 provide a summary of the
annual and seasonal characteristics of the near-surface stability and maximum stability below 500 m
across all sites. Figure 13 shows the frequency of the near-surface stability regime groupings (e.g., all
NN, regardless of aloft stability, all VSM, regardless of aloft stability, etc.) and the maximum stability
present in the entire profile, either near surface or above the boundary layer and below 500 m (e.g., the
frequency of the basic near-surface stability regime WS and all -WSA cases, all the MS and -MSA, cases,
etc.). Table S1 lists the frequency of WS and weaker, MS and stronger, and SS and stronger stability near
the surface and for the strongest stability below 500 m.
Somewhat surprisingly, regimes with near-surface stability WS and weaker (Table S1) are the
most common regimes at the interior sites in summer (63.2% of the time at South Pole and 61.4% of the
time at Dome C; Figure 13c, Table S1). However, this weaker near-surface stability is often capped by
stronger stability above the boundary layer, such that when considering the maximum stability below 500
m, regimes with stability MS and stronger occur 86.7% of the time at South Pole and 81.9% of the time at
Dome C. This indicates that moderate or stronger stability dominates aloft even though weaker stability
occurs most of the time near the surface in the summer This observation of enhanced stability above a
weakly stable boundary layer has not been widely documented, much less quantified, especially in the
continental interior of Antarctica. While winter at Dome C is characterized almost entirely by near-
surface stability regimes SS and stronger (96.9%), the winter at South Pole experiences these regimes less
often (68.8%; Figure 13g). However, when considering the maximum stability below 500 m (Figure 13h),
this reduced frequency of strong stability near the surface at South Pole compared to Dome C vanishes
and regimes with stability SS and stronger occur nearly continuously and with similar frequency at both
South Pole and Dome C (99.6% and 99.2% of the time, respectively; Table S1).
Across all three coastal sites, WS and weaker near surface stability occurs more than 50% of the
time in all seasons, except for Neumayer in winter (Table S1). In the summer WS and weaker near
surface stability is dominant, occurring 80.1% to 92.1% of the time (Figure 13c, Table S1). However, this
high frequency of WS or weaker stability near the surface is not evident when stability aloft is considered
and WS and weaker stability anywhere below 500 m occurs 42.1 to 59.6% of the time (Figure 13d, Table
S1). This indicates that while weaker near-surface stability is dominant in the summer at the coastal sites,
MS or stronger stability is nearly as frequent as WS or weaker stability above the boundary layer. In the
winter, WS and weaker near surface stability occurs 40% to 53.6% of the time (Figure 13g, Table S1)
indicating a near even split between near neutral to weak stability and moderate or stronger stability near
the surface. In contrast, MS and stronger stability is observed within the lowest 500 m 72.1% to 91.6% of
the time during the winter (Figure 13h, Table S1), indicating that weak near surface stability regimes
usually have enhanced (MS or stronger) stability aloft. At McMurdo, the existence of enhanced stability
above a layer of weaker stability was noted by Dice and Cassano (2022). Additionally, Silva et al. (2022)
described the boundary layer at Neumayer ranging from strong surface-based temperature inversions to
weak inversions near the surface with stronger inversions aloft throughout the year, which is also
observed here. While both Dice and Cassano (2022) and Silva et al. (2022) noted the presence of
enhanced stability above a layer of weaker stability, neither of these studies quantified the occurrence or
seasonality of this phenomenon.
Comparing the coastal to the continental sites, near-surface WS and weaker stability regimes are
much more common at the coastal sites (61.3% to 72.4%) compared to the continental interior sites
(13.5% to 27.2%) on an annual basis (Table S1). When considering the maximum stability below 500 m
MS and stronger stability occurs nearly all of the time at the interior sites (96.5 to 96.7% of the time) and



occurs more than half of the time at the coastal sites (56.5% to 76.6% of the time) annually (Table S1).
This is consistent with observations from Zhang et al. (2011) who found that surface-based temperature
inversions are less common along the coasts, as the coastal region is warmer, moister, and windier than
the continental interior, which all reduces near-surface stability.
In the summer, near-surface stability of WS or weaker occurs most of the time at all sites but is
more frequent at the coastal (80.1% to 92.1% of the time) compared to the continental sites (61.4% to
63.2% of the time) (Table S1). In comparison, near-surface stability regimes SS and stronger only occur
0.5% to 5.9% of the time at the coastal and 7.5% to 10.3% of the time at the interior sites, indicating the
rarity of strong near surface stability at both coastal and interior sites in the summer. However, when also
considering stability just above the boundary layer MS and stronger stability occurs more than 80% of the
time at both South Pole and Dome C (Table S1). Even at the coastal sites, MS and stronger stability
occurs nearly half of the time (40.4 to 57.9%) in the summer (Table S1). These results highlight that
while weak stability is usually present near the surface across the Antarctic continent in the summer,
moderate or stronger stability is often present somewhere in the lowest 500 m of the atmosphere.
In the winter, strong stability is expected to be dominant across Antarctica (Lettau and
Schwerdtfeger, 1967; Phillpot and Zillman, 1970; King and Turner, 1997; Andreas et al., 2000).
Surprisingly, the near-surface stability of WS and weaker still occurs 40.0% to 53.6% of the time in the
winter at the coastal sites, whereas these regimes, as expected, are infrequent at the interior sites,
occurring 14.1% of the time at South Pole and 0.8% of the time at Dome C (Figure 13g, Table S1). Near
surface stability stronger than SS occurs 12.3% to 33.4% of the time at the coastal sites and 68.8% to
96.9% of the time at the interior sites (Table S1), emphasizing the dominance of strong near surface
stability in the continental interior in winter. When considering the maximum stability below 500 m, it is
important to note that even though about half the time WS and weaker regimes occur near the surface at
the coastal sites, above the boundary layer enhanced stability remains. MS and stronger stability within
the lowest 500 m of the atmosphere occurs 72.1% to 91.6% of the time at the coastal sites. (Figure 13h,
Table S1). While there are very few cases with WS or weaker near surface stability at the continental
interior sites in the winter these always have enhanced stability above the boundary layer (Figure 13h).
The maximum stability below 500 m at the interior sites is almost always MS and stronger (99.8% to
100%), but in fact, the maximum stability is almost just as often SS or stronger (99.2% to 99.6%) (Table
S1). This emphasizes the near complete dominance of the SS, VSS, and ESS regimes in the continental
interior during the winter, while these regimes represent half or fewer (18.2% to 54.3%) of cases when
considering maximum stability below 500 m at the coastal sites in the winter (Figure 13h, Table S1).
It is also interesting to note the frequency of stability regimes in the spring and fall in comparison
to that in the summer and winter at all five sites. At the interior sites, there is a tendency for the regime
frequencies, whether considering just near surface stability or the maximum stability in the lowest 500 m,
in the fall and spring to mirror the winter season regime frequencies, and summer is completely distinct
from the other seasons (Figure 13c through 13j, Table S1). The most common near-surface stability
groupings in the fall and spring are WS and weaker at the coastal sites (55.7% to 71.8% of the time;
Figures 13e and 13i), and these regimes are observed less frequently in the transition seasons than they
are in the summer (80.1% to 92.1%; Figure 13c), but more frequently than in the winter (40% to 53.6%;
Figure 13g). In comparison, the transition seasons at the continental interior sites are usually characterized
by MS and stronger stability near the surface (77.7% to 95.4%; Figure 13f and 13j), which is similar to
the frequency of these regimes in the winter as well (85.8% to 99.5%; Figure 13g). Thus, at the interior
sites, this comparison emphasizes the quick descent into winter-like conditions in the transition seasons,
whereas at the coastal sites, this change is more gradual.



*Figure 13: Summary of the basic near-surface stability regime frequency (left column) and aloft stability regime frequency (right column) at all five sites annually (top row) and seasonally: summer, fall, winter and spring (bottom four rows). The colored bars indicate the frequency of each of the given regimes at each site: South Pole (dark blue), Dome C (light blue), McMurdo (yellow), Neumayer (orange), and Syowa (red).*



To assess how applicable the commonly cited clear, strongly stable and cloudy, weakly stable
description of polar winter boundary layers (Stone and Kahl, 1991; Mahrt et al., 1998; Mahrt, 2014;
Solomon et al., 2023) is for the Antarctic we applied the method of Solomon et al. (2023) to identify clear
and cloudy conditions, based on net longwave radiation. This approach for identifying clear and cloudy
conditions was successful at the coastal Antarctic sites (Figures S3 to S5) but was unable to identify
distinct radiative signatures for clear or cloudy conditions at the two interior sites (Figures S1 and S2).
This suggests there may be fundamental differences in processes related to clouds, radiation, and stability
on the polar plateau in comparison to the coastal region of Antarctica or over Arctic sea ice. Vignon et al.
(2017) suggested that there may be two distinct boundary layer regimes (weakly stable and strongly
stable) at Dome C, but contrary to locations in the Arctic (Solomon et al., 2017), this is likely due to a
critical shift in wind speeds, not a bimodal distribution in radiative forcing (Vignon et al., 2017).
For the three coastal sites, the frequency of the 20 boundary layer stability regimes defined in
Table 2 was calculated for clear and cloudy conditions (Figure 12). This analysis revealed MS and
stronger regimes occur more often with clear conditions rather than cloudy conditions at McMurdo and
Neumayer. The NN and WS regime grouping at McMurdo (excluding NN-SSA) and the NN regime
grouping at Neumayer occur more often with cloudy rather than clear conditions, but these are the only
stability regimes in this analysis in which there is a large difference in frequency for cloudy or clear
conditions. At Syowa, there is little difference in the frequency of any stability regime for both clear and
cloudy conditions. The fact that some stability regimes at McMurdo and Neumayer and all the stability
regimes at Syowa show little sensitivity to changes in cloud cover suggest a more nuanced relationship
between radiative forcing and near-surface stability may exist in the Antarctic compared to the Arctic, and
other forcing mechanisms, such as mechanical mixing, may be relatively more important in distinguishing
boundary-layer stability regimes from one another. Mahrt (2014) noted that weakly stable conditions
occur with either cloud cover or increased wind and mentioned that classification into the weakly stable
and strongly stable regimes does not encompass the full complexity of forcing in the stable boundary
layer.
A useful next step in this research will be to more thoroughly assess the forcing for the different
stability regimes. Largely, radiative forcing and mechanical mixing (wind shear) are two main drivers of
boundary layer stability. The role of these two processes, across seasons at the individual sites, but also
across the five sites will be the basis of continued research. Assessing forcing for regimes that showed
little sensitivity to cloud cover is of interest since it appears that changes in radiative forcing may not play
a dominant role. A paper following this study will use the boundary layer regimes identified for each
individual radiosonde profile to identify variations in radiation and wind speed associated with the
different stability regimes. Further, an analysis of the ability of the Antarctic Mesoscale Prediction
System (AMPS, Powers et al., 2012) to simulate the range of stability regimes observed at each site and
the radiative and mechanical forcing associated with these regimes across Antarctica is planned.



**Data Availability**

The data used to support this project can be found at:

McMurdo:

 All data: https://adc.arm.gov/discovery/#/results/site_code::awr.

Syowa:

 Radiosonde data: Office of Antarctic Observation Japan Meteorological Agency (pers. comm. Yutaka Ogawa)

 Radiation data: https://doi.pangaea.de/10.1594/PANGAEA.956748

Dome C:

 Radiosonde data: https://www.climantartide.it/dataaccess/rds/index.php?lang=it&rds=DOMEC

 Radiation data: https://doi.pangaea.de/10.1594/PANGAEA.935421

South Pole:

 Radiosonde data: http://amrc.ssec.wisc.edu/data/ftp/pub/southpole/radiosonde/

 Radiation data: https://doi.pangaea.de/10.1594/PANGAEA.956847

Neumayer:

 Radiosonde data: https://doi.org/10.1594/PANGAEA.940584

 Radiation data: https://doi.org/10.1594/PANGAEA.940584

**Competing Interests**

The contact author has declared that none of the authors has any competing interests.

**Acknowledgements**

Funding for this work came from the United States National Science Foundation (NSF) grant OPP 1745097 and the National Aeronautics and Space Administration (NASA; award 80NSSC19M0194).The authors thank the United States Antarctic Program, the Department of Energy, the Baseline Surface Radiation Network, the Antarctic Meteorological Research and Data Center, the Antarctic Meteo-Climatological Observatory, and the Office of Antarctic Observation Japan Meteorological Agency for the support and logistics for the data used in this paper.

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
