# Peer review of "Variations in Boundary Layer Stability Across Antarctica: A Comparison Between Coastal and Interior Sites"

_EGUsphere, 2023_

## Author Comment (AC1)

**Response to anonymous referee comments**

The authors thank the two referees for taking the time to review this manuscript and for their helpful comments, which have improved the manuscript. The text in the updated manuscript reflecting the changes made in this document is included below each response in quotation marks. Responses to referee comments are in *italics*. In areas where the response to the referee includes a specific edit or addition to text in the manuscript, these edits are noted in *highlighted yellow*.

**Reviewer 1: John King**

**General comments**

In this well-written paper, high vertical resolution radiosonde observations from five Antarctic stations are used to study the static stability of the lowest few hundred metres of the atmosphere. Self-organizing maps (SOMs) are used to define stability regimes for each station. The seasonal variability in the frequency of occurrence of the regimes is studied and the regime structure is compared between the different stations. As a SOM is calculated for each station individually, it is difficult to compare SOMs across stations so SOM nodes are assigned to stability regimes based on a two-layer classification. A proxy for cloud cover is used to investigate the influence of clouds on stability. Interesting contrasts are found between the two interior stations and the three coastal stations studied and there is some discussion of the reasons for these contrasts and for similarities/contrasts with measurements made in the Arctic.

The paper is probably the most comprehensive study of boundary-layer and lower atmosphere stability in the Antarctic that has been carried out to date. Using a SOM approach to define stability regimes is novel but I would have liked to see a bit more explanation of how the SOM analysis (section 2.2.1) was used to inform the development of the stability regime classification (section 2.2.2). At the moment they are presented separately and it's not entirely clear what additional value the SOM analysis brings to the study.

The paper has a strong climatological focus, with rather limited discussion on what factors drive the various stability regimes that are observed, apart from some rather detailed analysis on the impact of cloud cover. I realise (from statements in the conclusions section) that future papers will examine controls on stability regimes more deeply but it seems a little strange to examine one factor in detail in this first paper and not to discuss other factors, such as wind shear, that may be of equal or greater importance. Any future studies will also need to recognise that stability within and above the boundary layer are controlled by different mechanisms. Within the boundary layer the main controls on stability are surface energy balance and mechanical mixing while, above the boundary layer, radiative flux divergence and dynamical processes such as advection and subsidence may dominate. You will need to look at forcings in both of these regions to fully explain the results of the current study.

Overall, I would recommend this paper for publication in WCD after minor to moderate revision. Below, I set out the main points that I would like to see addressed in a revised version of the paper.

*Thank you very much for your thorough and positive comments.*

*The SOM method is first used in the manuscript to reveal the range of boundary layer stability present at five sites across the Antarctic continent. This method is used because the SOM, unlike other clustering or*

*pattern identification techniques (EOFs, PCA, k-means clustering, etc.) identifies the range of conditions present in a given set of training data without* a priori *information. This is a useful way to examine what the full range of stability is like in the boundary layer at multiple sites (with hundreds or thousands of individual profiles) and gives a continuum of stability across the radiosonde data used, rather than simply a few completely distinct patterns. From this range of stability identified by the SOM algorithm, the boundary layer stability definition scheme was developed, in concert with another project which aimed to gain similar information from an Arctic study. Thus, the SOM informed the authors about the types of stability profiles present in the data, which then informed the development of the boundary layer stability definition scheme.*

*After review of all the information learned from the analysis presented in this manuscript, which discusses the relative frequencies of the stability regimes annually and seasonally at five sites across Antarctica, as well as the information gathered during review of the forcing mechanisms which you mention, the authors decided to split all of this detailed information into two manuscripts. The cloud analysis was included in this manuscript as it was primarily used to assess different frequencies of stability regimes, consistent with the focus of this manuscript. As you've mentioned, another manuscript regarding the forcing mechanisms for the various boundary layer stability regimes has been written and is now in preprint also in Weather and Climate Dynamics ([https://doi.org/10.5194/egusphere-2023-2062](https://doi.org/10.5194/egusphere-2023-2062)) and builds on much of the information gathered in this first manuscript.*

**Specific comments**

The study is based on observations from radiosondes that are launched once or twice per day. During the Antarctic summer, there is a strong diurnal cycle of solar radiation at all of the stations studied, apart from South Pole. It is well-known that the diurnal cycle of solar radiation at Dome C during the summer strongly modulates the structure of the atmospheric boundary layer at that location (Mastrantonio et al, Meteorol. Atmos. Phys., 71, 127-132, 1999; King et al., 2006, doi: 10.1029/2005JD006130). The 1200 UTC daily sounding at Dome C takes place at around 0400 local time and is thus representative of early morning conditions, when the boundary layer is both shallower and more stably stratified than it is during the middle of the day. This is not a serious issue with the study but it should be mentioned in the methodology section and when the Dome C results are presented in section 3.2.

*Thank you for this important comment. The authors have added the following text (Lines 140-144 in CLEAN edited manuscript) in Section 2.1 in the paragraph about the Dome C information:*

> *"It is important to note here that the 1200 UTC soundings are 0400 local time, which is early morning at Dome C. Thus, at this time, the profiles from the radiosondes are likely to be reflective of shallower, more stable boundary layer conditions, rather than convective which is sometimes observed in near surface observations during mid-day or in the summer at Dome C (Mastrantonio, et al., 1999; Pietroni et al., 2013)."*

*Additional text which identifies the local times at the other four sites has also been added in Section 2.1.*

Section 3.6. The absence of a strong link between cloudiness and stability at the interior stations is not surprising. South Pole and Dome C are both characterised by a much greater frequency of cloud-free conditions than is typical of coastal Antarctic stations or Arctic Ocean locations. The clouds that do occur

at these high-altitude stations are often optically-thin as they contain little or no liquid water, so cloud classification based on downwelling longwave radiation may not work well for these stations and the impact of these clouds on surface energy balance (and hence on stability) is likely to be very different to that of optically-thick mixed-phase clouds over the Arctic Ocean and Antarctic coastal margins. See, e.g., Town et al., 2007, https://doi.org/10.1175/JCLI4005.1 , Ganeshan et al, 2022, doi:10.1029/2022JD036801

*The authors realize that the lack of distinction between a clear and cloudy state at the continental interior sites was mentioned (as estimated by PDFs of net longwave radiation), but a reason was not given and this is something that should be added. The information provided here was useful, and text was added to Section 3.6 to address this comment, following the original sentence in the manuscript which states "The overlap ratio for the cloudy and clear downwelling longwave radiation PDFs for each site, as described in Section 2.2.3, further support the lack of distinct cloudy and clear radiative states at the interior sites, with large values of this ratio (0.84 at South Pole and 0.91 at Dome C) indicating that there is no value of net longwave radiation that allows a meaningful separation between cloudy and clear states with unique distributions of downwelling longwave radiation." The following text was added (Lines 595-599 in CLEAN edited manuscript) immediately after this sentence:*

> *"The inability to find a distinction between the clear and cloudy states at the continental interior sites may be related to the fact that previous studies have noted that the cold, dry atmosphere of the continental interior of Antarctica is conducive to infrequent high, optically thin ice clouds, rather than optically thick liquid or mixed-phase clouds which are lower and have higher near-surface radiative impacts (Morely et al., 1989; Town et al., 2005, 2007; Ganeshan et al., 2022)."*

Lines 659-660: "Somewhat surprisingly…". Is this surprising? Strong near-surface stability in winter is the result of strong radiative cooling of the surface, which has to be compensated for by a large downward turbulent heat flux which drives strong stratification in the near-surface layer. During December and January at South Pole, net solar radiation almost exactly balances net longwave radiation at the surface so the turbulent heat flux (and, consequently, the near-surface temperature gradient) is small (King and Connolley, J. Climate, 1997, https://doi.org/10.1175/1520-0442(1997)010<1273:VOTSEB>2.0.CO;2)

*The authors wrote this initially as "somewhat surprisingly" given that several other studies have noted the near constant temperature inversion at continental interior locations throughout the year, even in the summer. For example, Hudson and Brandt (2005) found an inversion over half the time in summer at South Pole, and Genthon et al. (2013) noted an inversion 85% of the time at Dome C in the summer, which is only interrupted during the afternoon with convection. Hudson and Brandt (2005) noted that these inversions are usually less than 5 K (100 m)$^{-1}$ (consistent with our definition of the MS regime) but can sometimes reach 25 K (100 m)$^{-1}$ (between our definition of the SS and VSS regime), although their measurements extended from 2 m above the surface, where these gradients are often higher, and our measurements start at 20 m above the surface. Genthon et al. (2013) noted that often the inversions at Dome C in the summer are around 2 K (100 m)$^{-1}$ (between our definition of the WS and MS regime), extending from 3.3 m above the surface. Thus, it was surprising that, at the surface at South Pole and Dome C in our study that over 60% of the time WS and weaker (less than 1.75 K (100 m)$^{-1}$) stability was present, when other studies have pointed out the inversion conditions stronger than this are usually present even in the summer. While it is true that the inversions observed by Hudson and Brandt (2005)*

*and Genthon et al. (2013), among others, are weaker in the summer in comparison to the winter due to this lesser turbulent heat flux, an inversion is still often observed.*

*The following text has been added (Lines 685-688 in CLEAN edited manuscript) to the manuscript in the discussion section:*

> *"It has been previously described in the literature that, even during austral summer, a temperature inversion is present nearly constantly (Hudson and Brandt, 2005; Genthon et al., 2013). Other studies, however, note the possibility of unstable conditions in the summer (King and Connolley, 1997; Mastrantonio, et al., 1999; Pietroni et al., 2013). Thus, this study posed an opportunity to evaluate the range of stability present in the summer season across multiple Antarctic sites."*

*The "somewhat surprisingly" text has been removed.*

**Minor points and technical corrections**

Line 36: "coastal"

*Thank you, this change has been made.*

Lines 46-47: You could also add W. Connolley, 1996, Int. J. Climatol., 16, 1333-1342 as a more recent reference here.

*Thank you, this addition has been included.*

Lines 201-201: Is this a subjective judgement or were any objective criteria used?

*Several objective criteria were used, including the use of the Sammon Map, which shows how adjacent nodes are related to each other in the SOM (Cassano et al., 2015), as well as statistics relating the BMUs in each SOM node (pattern) to the SOM-identified profile, including root mean squared error, mean absolute error, bias, and correlation. These statistics, as well as the comparison of the general layout of the various SOM sizes tested (3 x 2 (6 patterns), 4 x 3 (12 patterns), 5 x 4 (20 patterns), 6 x 5 (30 patterns), and 7 x 6 (42 patterns)) revealed that both the 3 x 2 and 4 x 3 SOMs had too large of errors between the SOM identified-profiles and the observations, and did not capture many of the unique profiles that the larger SOMs did, which thus rendered them insufficient. The largest size (7 x 6) was simply too large, and patterns become too similar to each other to discern useful differences. Ultimately, the 6 x 5 size, with respect to all these metrics, was the best choice.*

Lines 239-241: Please include an equation that shows how you calculate the bulk Richardson number. Note that Rib is only an approximation for the ratio of buoyancy production/destruction to shear production.

*This equation has been added in Section 2.2.2, as well as the following text (in highlight within the original sentence below; Line 259 in CLEAN edited manuscript):*

*"The bulk Richardson number is defined as the ==approximation of the== ratio of buoyant turbulence production, or suppression, to mechanical generation of turbulence by wind shear."*

Lines 335-336: "...or in the Arctic"? Maybe you should also make it clear that you are talking about the Arctic Ocean here – very strong stability is seen in Arctic regions such as Siberia or over the Greenland ice sheet.

*This edit has been made to say "or" instead of "even". This clarification has also been made about the "Central Arctic", as below (Line 358 in CLEAN edited manuscript):*

> *"...are rarely observed outside of the interior of Antarctica, ==or== over the Greenland ice sheet or Siberia in the Arctic (Zhang et al., 2011)"*

Figure 2: I think that the top and bottom x-axis descriptions are swapped round in the figure caption.

*The figure captions have been corrected.*

Figure 7: Please label each panel as you have done in figure 5.

*This has been corrected.*

Lines 601-603: Mixing in strong winds associated with coastal cyclones is probably the dominant control on stability at Syowa.

*Yes, this will be further discussed in the manuscript which will follow this one, regarding the forcing mechanisms for varying stability across the continent.*

Lines 638 and 639: "many of the SOM profiles" rather than "much of the SOMs"?

*This edit has been made, as per below (Line 664 in CLEAN edited manuscript):*

> *"...but across ==many of the SOM profiles== for McMurdo (Figure 6) and Neumayer (Figure 8), and some of the SOM profiles for Syowa as well (Figure 10)."*

Line 738: The "...quick descent into winter-like conditions in the transition seasons" is often referred to as the "coreless winter".

*This sentence has been edited to reflect this useful comment (Line 769 in CLEAN edited manuscript):*

> *"Thus, at the interior sites, this comparison emphasizes the quick descent into the ==coreless winter from== the transition seasons, whereas at the coastal sites, this change is more gradual."*

**Reviewer 2: Anonymous Referee**

**General**

This paper analysis a large number of radiosoundings launched at five Antarctic stations, two continental ones and three coastal ones. The authors propose a classification of stability including near-surface conditions and conditions aloft. They find a wide range of potential temperature profiles and profiles of gradients, for which they propose 30 boundary layer regimes using the self-organizing maps neural network algorithm (SOM). They find large differences between the coastal and continental sites and finally distinguish cloudy and clear-sky regimes.

In most parts the paper is clearly written and to my knowledge a similar work comparing soundings of several Antarctic stations is not yet available. In principle, I like the work and recommend its publication

but at some points I have difficulties to follow and I think that the description could be clearer in some aspects explained below.

*Thank you very much for the positive comments. We look forward to addressing your concerns below.*

**Major Revisions**

1) Previous work (e.g. Handorf et al.) has shown that the Antarctic boundary layer can be extremely shallow with tops below 25 m height (sometimes 10-20 m, see their Figure 1). It is a challenge to measure such boundary layers by radiosoundings. Note that at Neumayer, soundings are launched from the station roof at 28 m above the surface. Also for other stations, the given lowest measurements at 20 m are not really 'near-surface'. The real ABL might be below, which has a large impact on turbulent fluxes. This should be explained.

*Thank you for this comment. Of course, this is a challenge with radiosondes. A sentence has been added in Section 2.2.1 to address this (Lines 198-203 in CLEAN edited manuscript):*

> *"It is also important to note here that the boundary layer in Antarctica has been observed to be shallower, and stable conditions extend further to the surface, than the 20 m bottom height in the profiles used in this analysis (e.g., Handorf et al., 1999). However, below this height in the radiosonde profiles, anomalously warm biased temperatures are important to exclude, since this will indicate weaker stability than are actually present during the radiosonde launches."*

2) Perhaps I was too fast, but it is difficult to understand that in Figures 2,4,6,8,10 the number of regimes amounts to 30 and thus differs from the number 20 in Table 3. Also, I cannot really follow why, e.g. in Figure 4 the same name SS occurs for classes 17 and 23.

*The SOM has 30 nodes and was used as an initial way to visualize the full range of stability profiles present in thousands of radiosonde soundings. Based on this visualization of the data we defined regimes based on vertical potential temperature gradient thresholds that allow for direct comparison across multiple sites (since unique SOMs were used to visualize the profiles at each site). Therefore, not each SOM node equates to one regime. As you have noted in Figures 2, 4, 6, 8, and 10, some SOM patterns have the same boundary layer regime since slightly different stability profiles can result in the same regime classification. The boundary layer stability regime definitions are then applied to the SOM figures (2, 4, 6, 8, and 10) as well as each individual radiosonde profile. The application of the regime definitions to the SOM provides a way for the reader to visualize what each stability regime looks like. The following text to clarify this has been added in Section 2.2.2 (Lines 282-285 in CLEAN edited manuscript):*

> *"The boundary layer stability regimes defined here are then applied to the patterns in the SOMs to show how this definition scheme applies to the range of potential temperature gradient profiles originally identified in the SOM, which was used to inform the development of the boundary layer stability regime definitions."*

3) Lines 220-224: As far as I understand the classification in Figures 2,4,6,8,10 is different for each station with different patterns. I did not understand why not a general classification is possible being valid for all

stations. In the present form, an intercomparison of results for different stations becomes difficult. Also, it would become difficult to see if results of a model would fit into one of the different classes. I think this requires more explanation.

*The SOM is used as a way to visualize thousands of profiles at each site in a compact way. Comparing the SOM figures (2, 4, 6, 8 and 10) one notes that the details of the profiles differ markedly across the five study sites. Therefore, using a single SOM would mask important differences in the data from each site and thus a SOM is created for each individual station. The results from the SOM analysis was then used to inform the development of a boundary layer definition scheme which is applicable to all sites across the continent. The following text has been added (in yellow highlight) in Section 2.2.2 to clarify this further (Lines 234-240 in CLEAN edited manuscript):*

> *"However, this analysis does not allow for direct, quantitative comparison across the five sites since unique SOMs are defined for each location, and the results below will show that the range of stability at each of the five sites is very different. Thus, to compare the range of boundary layer stability present at each of the five sites (Figure 1) the potential temperature gradient profiles, as shown by the SOMs at each of the study sites, are used to define boundary layer stability regimes that can be applied across all of the sites. "*

4) The present work gives the impression that the Antarctic boundary layer is always near-neutral or stably stratified. It should be stressed that near-neutral could also include convective cases. Note that, e.g. at Kohnen station a daily cycle is often observed with upward fluxes of sensible heat and thus convective conditions during daytime in a shallow boundary layer (e.g. Van As et al., 2005).

*Thank you for this comment, this is in fact evident in the SOM analysis, as several of the SOM patterns shown at Syowa and McMurdo have slightly negative potential temperature gradients near the surface. This has been noted in Section 2.2.2, with the newly added text (Lines 249-253 in CLEAN edited manuscript):*

> *"It is also important to note that the NN regime with potential temperature gradients less than 0.5 K (100 m)$^{-1}$ may include some negative potential temperature gradients, thus convective conditions, which, while rare in the Antarctic, can occur with strong radiative heating during the austral summer, or advection of cold air over a relatively warmer surface."*

5) The authors do not consider the effect of condensation on the stability. So, what is called weakly stable might be convective (or near-neutral) when the equivalent potential temperature is considered, so that the presented findings might be missleading in some sense. I recommend that this is explained.

*While equivalent potential temperature is important to consider in environments with regular moist convection, we feel that the cold, dry state that dominates the Antarctic, even in the summer, and the near complete lack of deep, moist convection does not warrant the use of equivalent potential temperature for this analysis.*

**Minor revisions**

Lines 86-88: There is an effect of clouds which has a strong impact on the shape of the potential temperature profile and thus on stability, which is not mentioned in the paper. This is the cloud radiative forcing and subsequent mixing (see, e.g. Chechin et al., 2023). Perhaps, it can be added here.

*Thank you for this comment. The following has been added to the text (Lines 88-89 in CLEAN edited manuscript):*

> *"Cloudy conditions can also result in cloud-top radiative cooling and initiate convective mixing when the atmosphere is cooled aloft by the cloud (Chechin et al., 2023)."*

Line 48: Usually near-neutral (throughout the paper).

*This has been updated throughout the paper to read as "near neutral" (without dash).*

Line 615: replace present by presented (?)

*We believe this change is not appropriate. This sentence reads as that these stability regimes are "present" (e.g., "are observed") not "presented".*

Lines 796 and 797: When you click onto the given PANGAEA links, you can find the sentence: Always quote citation above when using the data! This means here that the correct citation (which should occur in the list of references) is: Schmithüsen, Holger (2022): Radiosonde measurements from Neumayer Station (1983-02 et seq). Alfred Wegener Institute, Helmholtz Centre for Polar and Marine Research, Bremerhaven, PANGAEA, https://doi.org/10.1594/PANGAEA.940584

This might be similar for the other data sources.

*Thank you. These citations have been added to the references list and we have verified that all other data is cited correctly.*

Line 23, 62 and many other places: The present name of the station is Neumayer Station III, and before that it was called Neumayer Station. Only the first station was called Georg von Neumayer Station. One should write simply Neumayer Station when all stations are addressed.

*Line 62 in the original manuscript specifies that Neumayer will be listed as simply "Neumayer":*

> *"Georg von Neumayer Station III (Neumayer Station)"*

*The only other place this station was referred to as "Georg von Neumayer" was in Table 1 which has now been updated to read as simply "Neumayer".*

---

## Author Response (AR2)

Equation (1): Please define all of the variables in this equation and state the value of delta-z used in the calculation of the bulk Richardson number.

*Author comment: Please see lines 264-266 in newly revised manuscript which defines these quantities.*